# PRACTICAL AND PRIVATE HETEROGENEOUS FEDERATED LEARNING

## ABSTRACT

Heterogeneous federated learning (HFL) enables clients with different computation/communication capabilities to collaboratively train their own customized models, in which the knowledge of models is shared via clients' predictions on an auxiliary unlabeled dataset. However, there are two major limitations: 1) The assumption of auxiliary datasets may be unrealistic for data-critical scenarios such as Healthcare and Finance. 2) HFL is vulnerable to various privacy violations since the samples and predictions are completely exposed to adversaries. In this work, we develop PrivHFL, a general and practical framework for privacy-preserving HFL. We bypass the limitation of auxiliary datasets by designing a simple yet effective dataset expansion method. The main insight is that expanded data could provide good coverage of natural distributions, which is conducive to the sharing of model knowledge. To further tackle the privacy issue, we exploit the lightweight additive secret sharing technique to construct a series of tailored cryptographic protocols for key building blocks, such as secure prediction. Our protocols implement ciphertext operations through simple vectorized computations, which are friendly with GPUs and can be processed by highly-optimized CUDA kernels. Extensive evaluations demonstrate that PrivHFL outperforms prior art up to two orders of magnitude in efficiency and realizes significant accuracy gains on top of the stand-alone method.

## 1 INTRODUCTION

Heterogeneous federated learning (HFL) (Li & Wang, 2019; Chang et al., 2019), as a promising variant of federated learning (FL), enables clients equipped with different computation and communication capabilities to collaboratively train their own customized models that may differ in size, numerical precision or structure (Lin et al., 2020). In particular, clients share the knowledge of models via their predictions on auxiliary datasets, such as unlabeled problem domain datasets (Choquette-Choo et al., 2021) and public non-problem domain datasets (Li & Wang, 2019; Lin et al., 2020). This flexible approach facilitates customized FL-driven services in areas like Healthcare and Finance (Kairouz et al., 2019), while solving the intellectual property concerns of FL models (Atli et al., 2020). However, HFL suffers from two major limitations: (1) The assumption of auxiliary datasets may be unrealistic for many data-critical scenarios (Zhu et al., 2021). For example, in Healthcare applications, task-related auxiliary datasets that contain patients' sensitive information are usually difficult to obtain due to current strict regulations like General Data Protection Regulation. (2) Sharing predictions may still leak the privacy of local data (Papernot et al., 2017). Several works have demonstrated that given the black-box access to a trained model, adversaries can infer membership (Salem et al., 2019) and attribute information (Ganju et al., 2018) of the target sample, and even can reconstruct the original training data (Yang et al., 2019). Therefore, to promote the deployment of HFL in real-world applications, it is crucial to solve the above two problems.

To the best of our knowledge, in HFL the relaxation of the auxiliary dataset assumption has not been explored before. Specifically, it is challenging to achieve collaborative training under heterogeneous models when there is no an auxiliary dataset as a medium for the model knowledge transfer (Li & Wang, 2019). On the other hand, to mitigate the above privacy risks, a natural solution is to integrate advanced secure prediction protocols, such as CrypTFlow2 (Rathee et al., 2020), CryptGPU (Tan et al., 2021), and HE-transformer (Boemer et al., 2019b). These schemes can protect the private information during the model knowledge transfer by utilizing homomorphic encryption (HE) (Gentry,

Under review as a conference paper at ICLR 2022

2009), garbled circuit (GC) (Yao, 1986) or oblivious transfer (OT) (Naor & Pinkas, 2001) techniques (refer to Appendix D.2 for more details). Unfortunately, such methods add huge computation and communication overhead due to the use of heavy cryptographic primitives. For instance, Choquette-Choo et al. (2021) recently proposed CaPC, the first private collaborative learning scheme based on HE-transformer (Boemer et al., 2019b) supporting heterogeneous models, which can be directly extended to HFL. As mentioned above, this work still suffers from efficiency issues, inherited from prior secure prediction protocols. Moreover, their scheme is implemented by the interaction between clients[1], however in real-world applications, clients (e.g., mobile devices) generally cannot establish direct communication channels with others (Bonawitz et al., 2017). Therefore, the challenge here is how to efficiently implement secure prediction protocols in the realistic HFL setting.

In this work, to approach the above challenges, we develop PrivHFL, a general and practical framework for privacy-preserving HFL. First, PrivHFL relaxes the assumption of dependence on auxiliary datasets and designs a simple but effective dataset expansion method only with clients' private datasets. To this end, we instantiate it by leveraging mixup (Zhang et al., 2018) that is originally a regularization technique to improve generalization, and also present some exploration with other data augmentation methods like cutout (DeVries & Taylor, 2017) and cutmix (Yun et al., 2019). The key idea is that the expanded data could provide good coverage of natural dataset distributions and hence could be used as an effective medium for transferring model knowledge. Second, to securely and efficiently evaluate HFL, we leverage the lightweight additive secret sharing technique (Demmler et al., 2015) to construct customized secure prediction protocols *from scratch* in a practical setting where there is no direct communication between clients. Our gains mainly come from the improvement in communication and computation through the elimination of costly HE and GC protocols. Moreover, in contrast to prior works that evaluate cryptographic protocols in CPUs, PrivHFL converts complex cryptographic operations to simple computations on large blocks of data, which are friendly with GPUs and can be processed by highly-optimized CUDA kernels (Tan et al., 2021). As a result, PrivHFL is suitable for the batch prediction (i.e., performing multiple predictions at the same time) with lower amortized cost. We evaluate the designed protocol on GPUs and CPUs, and the results show that our GPU-based protocol is up to $10\times$ faster than its CPU analog. Our key contributions can be summarized as follows:

- We introduce a practical HFL framework, which is independent on any auxiliary datasets while provably providing comprehensive privacy protection.
- We design a simple yet effective dataset expansion method to promote the sharing of model knowledge, and construct customized cryptographic protocols for secure prediction.
- Extensive experiments on SVHN, CIFAR10, Tiny ImageNet (including IID and Non-IID settings) and various heterogeneous models demonstrate that PrivHFL outperforms prior art up to two orders of magnitude in efficiency and realizes roughly 10% accuracy gains.

## 2 BACKGROUND

Before introducing PrivHFL, we first describe the heterogeneous federated learning and the threat model, and then review the cryptographic primitives that are required to understand our work.

### 2.1 HETEROGENEOUS FEDERATED LEARNING

In HFL (Li & Wang, 2019; Choquette-Choo et al., 2021), the clients independently design their own unique models, but due to the model heterogeneity, they cannot directly share model parameters with each other. Instead, they learn the knowledge of other models via the predictions on a task-related auxiliary dataset, where a server routes messages between the clients since they generally cannot establish direct communication channels with others (Bonawitz et al., 2017; Bell et al., 2020). Specifically, clients first train local models with their own private datasets. Then, each client performs prediction on the auxiliary dataset based on the local model and sends the prediction results to the server to aggregate. Later, the server broadcasts aggregated results to clients, which will retrain local models based on the auxiliary dataset and received predictions. The whole process is

---

[1]As shown in C.5, by carefully designing protocols, CaPC can be extended to the communication-limited setting but at the cost of increased communication overhead.

2

iterative until each local model meets the pre-defined accuracy requirement. Details can be referred to Figure 9 and Appendix A.

## 2.2 THREAT MODEL

We work in an honest-but-curious adversary setting (Goldreich, 2009), where each entity (including the clients and the server) strictly follows the specification of the designed protocol but attempts to infer more knowledge about other clients' private information such as model parameters and private datasets. Moreover, to maintain the reputation and provide more services, the server does not collude with any clients. Formally, an attacker either corrupts the server or a subset of clients but not both. This setting is reasonable and has been widely instantiated in previous works (Phong et al., 2018; Sun & Lyu, 2021; Choquette-Choo et al., 2021).

## 2.3 CRYPTOGRAPHIC PRIMITIVES

*Additive secret sharing.* We adopt lightweight 2-out-of-2 additive secret sharing over the ring $\mathbb{Z}_L$ (Demmler et al., 2015) as the cryptographic building block. We let $\mathsf{Share}(x)$ denote the sharing algorithm that takes as input $x$ in $\mathbb{Z}_L$ and outputs random sampled shares $[x]_0, [x]_1$ with the constraint $x = [x]_0 + [x]_1$ in $\mathbb{Z}_L$. Arithmetic operations can be implemented in the sharing form as shown in Appendix C.4.1. The reconstruction algorithm $\mathsf{Recon}([x]_0, [x]_1)$ takes as input the two shares and outputs $x = [x]_0 + [x]_1$ in $\mathbb{Z}_L$. The security of the additive secret sharing protocol guarantees that given a share $[x]_0$ or $[x]_1$, the value $x$ is perfectly hidden.

*Pseudorandom generator.* A Pseudorandom Generator (PRG) takes as input a uniformly random seed and a security parameter $\kappa$, and outputs a long pseudorandom string. The security of PRG ensures that the output is indistinguishable from the uniform distribution. In PrivHFL, PRGs enable two parties to generate same (pseudo-) random numbers without communication. We instantiate PRG with the technique from (Matyas, 1985) and the seed can be generated by the Diffie-Hellman Key Agreement protocol (Diffie & Hellman, 1976). Details can be referred to Appendix C.4.2.

## 3 THE PRIVHFL PROTOCOL

In this section, we introduce the high-level view of PrivHFL, followed by describing in detail our dataset expansion method and secure prediction scheme.

### 3.1 HIGH-LEVEL VIEW OF PRIVHFL

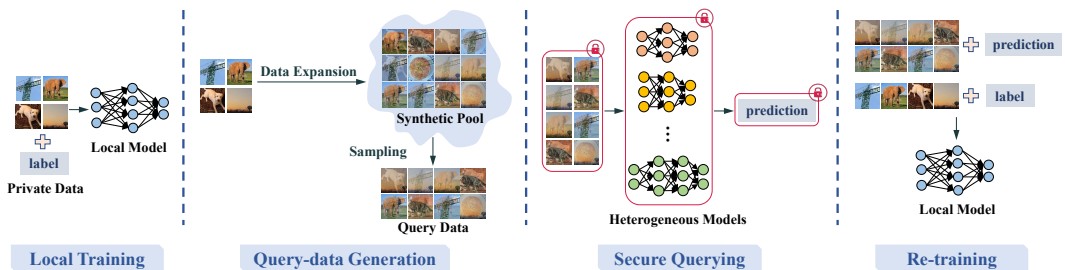

Figure 1: High-level view of PrivHFL

PrivHFL follows prior HFL works (Li & Wang, 2019) and iteratively optimizes the clients' models. Each client in PrivHFL can play the role of the querying party and the answering party at the same time, and without loss of generality, we denote them as $P_Q$ and $P_A$, respectively. As shown in Figure 1, in each iteration, each $P_Q$ performs four-phase operations with other $P_A$, i.e., *local training*, *query-data generation*, *secure querying*, and *re-training*. In detail, clients first train the local model on their own private datasets, which is the *baseline* any future improvements will be compared with. After that, by utilizing our dataset expansion method, each $P_Q$ can generate the query data to query other $P_A$ ($C$ fraction of all clients) for prediction results. To protect the privacy of query

samples, predictions and model parameters, clients adopt our secure prediction protocol and conduct collaborative querying in the ciphertext form. To the end, each client can retrain the local model based on the private dataset, as well as the query samples and corresponding prediction results. Algorithm 1 in Appendix C.1 gives the detailed description of PrivHFL.

## 3.2 QUERY-DATA GENERATION

To relax the assumption of auxiliary datasets, we design and instantiate an effective dataset expansion method, inspired by the success of mixup (Zhang et al., 2018) in improving model generalization and the efficiency of knowledge distillation (Wang et al., 2020). Specifically, we repurpose mixup to construct a big synthesized pool on the small private dataset, which could provide a good coverage of the manifold of natural samples. Given any two private samples $x_i$ and $x_j$, we generate multiple synthetic query samples by a convex combination with different coefficients $\lambda$ as follows:

$$\tilde{x}_{i,j}(\lambda) = \lambda \cdot x_i + (1 - \lambda) \cdot x_j. \tag{1}$$

Empirically, we set $\lambda \in [0.1, 0.9]$ with an interval of 0.1 to generate more diverse synthetic images. We also explore the influence of different $\lambda$ values in Appendix B.3 and Table 5. This simple method can exponentially expand the size of initial dataset and hence provide more candidate samples for query. Following CaPC (Choquette-Choo et al., 2021), we use random sampling and active learning strategies (Tong & Koller, 2001) in Appendix C.3 to select informative samples from the synthesized pool. Note that an alternative solution is to directly use private datasets to query like knowledge distillation (Hinton et al., 2015). We compare against this method in Section 4.

**Extensions of mixup-based method.** Dataset expansion based on private samples is a universal and modular method, therefore it can be readily extended with techniques in data augmentation literature. For example, one may replace the mixup-based dataset expansion in PrivHFL with recent methods such as cutout (DeVries & Taylor, 2017) and cutmix (Yun et al., 2019). We present some exploration and experiments with these extensions in Appendix B.1 and Figure 11.

## 3.3 SECURE QUERYING PROTOCOL

As shown in Appendix G and Figure 19, it is entirely possible to reconstruct the original image from the mixup-synthetic samples. To further protect the privacy of query samples as well as predictions and model parameters, we design a secure querying protocol by exploiting the lightweight additive secret sharing technique, rather than heavy HE and GC techniques. Recall that in our querying protocol, three entities are included, i.e., the server, $P_Q$ and $P_A$. The challenge here is how to efficiently implement such protocol under strict communication constraints, i.e., the communication channels cannot be established among clients. To tackle this challenge, the key idea is to outsource the secure querying task to the server and $P_A$, and then design a $P_Q$-assisted customized protocol to accelerate the evaluation. Moreover, an important design principle is to conduct a GPU-friendly protocol that mainly includes vectorized operations and hence is suitable for batch prediction with better amortized costs. In the following, we decompose our scheme into three steps: *query-data sharing*, *secure prediction*, and *result aggregation*.

**Query-data sharing.** We first construct PRG seeds in pairs for $P_Q$, $P_A$ and the server, denoted as $Sk_{QA}$, $Sk_{SA}$, and $Sk_{SQ}$, which are used to generate same random numbers without communication (refer to Figure 18 and Appendix C.4.1). Figure 2 shows our query-data sharing protocol $\Pi_{\text{Share}}$, in which $P_Q$ secret-shares the query data $x$ to the server and $P_A$ for secure prediction. In particular, $P_Q$ non-interactively shares $[x]_0 = r$ with $P_A$ using PRGs on the same seed $Sk_{QA}$. After that, $P_Q$ computes and sends $[x]_1 = x - r$ to the server.

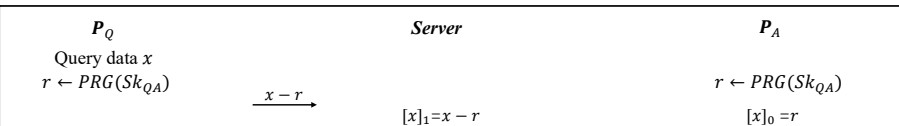

Figure 2: Secure query-data sharing protocol $\Pi_{\text{Share}}$

**Secure prediction.** Recently, many works achieve secure prediction protocols in client-server setting (i.e., 2-party setting), such as the SOTA CrypTFlow2 (Rathee et al., 2020) or HE-Transformer

(Boemer et al., 2019a) adopted by CaPC (Choquette-Choo et al., 2021). However, such methods add a huge overhead due to the use of heavy cryptographic primitives, e.g., HE and GC. The most efficient secure prediction protocol by far is CryptGPU (Tan et al., 2021) under the 3-party setting, but it cannot be directly applied to HFL due to the inability to communicate between clients. To achieve the efficiency of secure prediction and adapt to the communication-limited scenarios, we design customized protocols for the linear layers and non-linear layers[2] *from scratch*. Figure 17 in Appendix C.2 gives a graphic depiction of end-to-end secure prediction. Below, we elaborate on the evaluation of the linear layers, ReLU and MaxPooling.

| $P_Q$ | Server | $P_A$ |
|---|---|---|
| | $[x]_1$ | Model parameter $\omega$, $[x]_0$ |
| $a, [c]_0 \leftarrow PRG(Sk_{QA})$ | | $a, [c]_0 \leftarrow PRG(Sk_{QA})$ |
| $b \leftarrow PRG(Sk_{SQ})$ | $b \leftarrow PRG(Sk_{SQ})$ | $\xleftarrow{\omega + a}$ |
| $\xrightarrow{[c]_1 = ab - [c]_0}$ | | $\xrightarrow{[x]_1 - b}$ |
| | $[y]_1 = (\omega + a)b - [c]_1$ | $[y]_0 = \omega[x]_0 + \omega([x]_1 - b) - [c]_0$ |

Figure 3: Secure matrix multiplication protocol $\Pi_{\text{Matmul}}$

*(i) Linear layers.* Linear layers' evaluation follows the idea of the Beaver's multiplication in Appendix C.4.1, but we improve the communication efficiency using PRGs. Specifically, $P_A$ and the server compute matrix multiplication $\omega x$, where the model parameter $\omega$ is held by $P_A$ and the input $x$ is secret-shared between $P_A$ and the server. Given that $\omega x = \omega[x]_0 + \omega[x]_1$, $P_A$ can compute $\omega[x]_0$ locally. As shown in Figure 3, we design an efficient protocol $\Pi_{\text{Matmul}}$ for evaluating $\omega[x]_1$. In particular, $P_Q$ first generates three random matrices $a$, $b$ and $[c]_0$ using PRGs, and computes $[c]_1$ that satisfies $[c]_1 + [c]_0 = a \cdot b$. Accordingly, the server can generate the same $b$ using PRGs based on $Sk_{SQ}$, and $P_A$ obtains $a$ and $[c]_0$ using PRGs based on $Sk_{QA}$. $(a, b, c)$ is the Beaver's triple with the constrain $c = ab$ to mask the inputs of $P_A$ and the server. Later, the server and $P_A$ call the Beaver's multiplication protocol in Appendix C.4.1 to jointly compute $[y]_1$ and $[y]_0$, i.e., the secret shares of $\omega x$. Note that the evaluation of fully-connected, convolutional, and AvgPooling layers can be derived directly from $\Pi_{\text{Matmul}}$ (Wagh et al., 2019; Rathee et al., 2020).

*(ii) ReLU.* The formula of ReLU is $\text{ReLU}(x) = x \cdot \text{DReLU}(x)$, where $\text{DReLU}(x) = 1 - \text{sign}(x)$[3], namely that it equals 1 if $x \geq 0$ and 0 otherwise. Since the multiplication of $x$ and $\text{DReLU}(x)$ can be implemented by the protocol $\Pi_{\text{Matmul}}$, we mainly focus on the evaluation of $\text{DReLU}(x)$, i.e., $\text{sign}(x)$. The insight is to convert the calculation of $\text{sign}(x)$ to the calculation of $\text{sign}(r \cdot x)$ where $r$ is a random positive number. Based on the above observation, we design a secure DReLU protocol $\Pi_{\text{DReLU}}$ as shown in Figure 4. In detail, $P_A$ and the server first generate a random positive value $r$ using PRGs with $Sk_{SA}$, and compute $[z] = r[x]$ locally. Then they send the secret-shared value $[z]$ to $P_Q$. Note that since $r$ is randomly generated, the value of $r \cdot x$ is also a random number in the domain. Similar ideas are also applied to other privacy-preserving machine learning works (Wagh et al., 2019; Shen et al., 2020). After that, $P_Q$ computes the sign of $z$, i.e., the sign of $x$. $\text{sign}(z)$ is shared to the server and $P_A$ based on protocol $\Pi_{\text{Share}}$ in Figure 2.

| $P_Q$ | Server | $P_A$ |
|---|---|---|
| | $[x]_1$ | $[x]_0$ |
| | $r \leftarrow PRG(Sk_{SA})$ | $r \leftarrow PRG(Sk_{SA})$ |
| | $[z]_1 = r[x]_1$ | $[z]_0 = r[x]_0$ |
| $\delta, \delta' \leftarrow PRG(Sk_{QA})$ | | $\delta, \delta' \leftarrow PRG(Sk_{QA})$ |
| $\xleftarrow{[z]_0 - \delta + [z]_1}$ | | $\xleftarrow{[z]_0 - \delta}$ |
| $z = [z]_0 - \delta + [z]_1 + \delta$ | | |
| $\xrightarrow{sign(z) - \delta'}$ | | |
| | $[y]_1 = sign(z) - \delta'$ | $[y]_0 = \delta'$ |

Figure 4: Secure DReLU protocol $\Pi_{\text{DReLU}}$

---

[2]Deep learning models consist of a sequence of linear layers (e.g., fully-connected layers, convolutional layers and AvgPooling) and non-linear layers (e.g., ReLU and MaxPooling).

[3]The $\text{sign}(x)$ function is the most significant bit (MSB) of the value $x$.

*(iii) Maxpooling.* Maxpooling on $m$ values is computed by using a tree-reduction algorithm, which recursively partitions the input into two halves and then compares the elements of each half. Specifically, clients arrange the $m$ values into a 2-ary tree with depth $\log_2 m$, and evaluate the tree in a top-down fashion. Compared with the method in Rathee et al. (2020) that requires $m - 1$ communication rounds, our method achieves lower communication rounds with comparable computational overhead. In each comparison of two secret-shared element $[x]$ and $[y]$, we reduce it to the evaluation of ReLU. We observe $\mathsf{max}([x], [y]) = \mathsf{ReLU}([x] - [y]) + [y]$, and hence the computational complexity of Maxpooling evaluation mainly comes from the evaluation of $m - 1$ ReLU.

**Result Aggregation.** After the secure prediction, the server and each answering party $P_A^j$ hold the shares of predicted logits $[x_j], j \in [n]$, which will be aggregated and then returned to $P_Q$. As shown in Figure 5, $P_A^j$ and $P_Q$ first generate a random value $r_j$ based on PRGs. Then each $P_A^j$ computes $[x_j]_0 - r_j$ and sends it to the server. The server aggregates all received values and sends $\sum_{j=1}^{n} \left([x_j]_0 - r_j + [x_j]_1\right)$ to $P_Q$. Thus, $P_Q$ can reconstruct the aggregated logit $y = \sum_{j=1}^{n} x_j$ and obtains the soft label of the query data.

| $\boldsymbol{P_Q}$ | *Server* | $\boldsymbol{P_{A}^{j}}, j \in [n]$ |
|---|---|---|
| | $[x_j]_1, j \in [n]$ | $[x_j]_0$ |
| $r_j \leftarrow PRG(Sk_{QA})$ | | $r_j \leftarrow PRG(Sk_{QA})$ |
| | $\xleftarrow{\sum_{j=1}^{n}([x_j]_0 - r_j + [x_j]_1)}$ $\qquad\qquad \xleftarrow{[x_j]_0 - r_j}$ | |
| $y = \sum_{j=1}^{n}([x_j]_0 - r_j + [x_j]_1) + \sum_{j=1}^{n} r_j$ | | |
| $y \leftarrow softmax(y)$ | | |

Figure 5: Secure result aggregation protocol $\Pi_{\mathrm{Agg}}$

**GPU Acceleration.** Our protocols mainly consists of GPU-friendly vectorized secret-sharing, which can be processed by highly-optimized CUDA kernels. However, for multiplication operations, existing CUDA kernels are designed to operate on floating-point inputs. In PrivHFL, we typically compute over integer values. To leverage optimized kernels for protocol acceleration, our PrivHFL integrates the CUDALongTensor abstract in CryptGPU (Tan et al., 2021) that embeds the integer-valued cryptographic operations into floating-point arithmetic (refer to Appendix C.6 for more details). However, CryptGPU's protocols cannot be directly extended to PrivHFL in the communication-limited scenario, and hence we redesign all cryptographic protocols from scratch.

**Security analysis of PrivHFL.** We give a formal security proof in Appendix E. Intuitively, PrivHFL reveals zero information to $P_A$ and the server, and only reveals the final aggregated prediction to $P_Q$, since all intermediate values are secret-shared[4]. Given the above, a corrupted $P_A$ cannot learn anything about the query data of querying parties, while the confidentiality of answering parties' model parameters against corrupted $P_Q$ is also protected.

### 3.4 DISCUSSIONS

**Discussions on differential privacy extension.** The differential privacy (DP) guarantee can complement PrivHFL. A well-designed DP mechanism can be used as a plug-and-play module to prevent privacy leakage from the aggregated result. While several works have been proposed in the deep learning domain (Papernot et al., 2017; Sun & Lyu, 2021; Choquette-Choo et al., 2021), this is non-trivial to design a customized DP mechanism for HFL, because the privacy-utility tradeoff is difficult to resolve. Especially, the privacy guarantee will deteriorate with the increase of corrupted clients, unless it is mitigated by adding more DP noises at the cost of accuracy. In our setting, we assume up to $n - 1$ ($n$ is the number of clients) clients can be corrupted such that the above problem will be escalated to the worst case. Therefore, it is an interesting and challenging work to design a high-utility DP mechanism in the distributed scenario where multiple clients may be corrupted.

**Discussions on scalability.** In PrivHFL and general heterogeneous federated learning, each client can play the role of the querying party and the answering party at the same time (Li & Wang, 2019). This will incur the overhead of $O(n^2)$ secure predictions for each iteration, where $n$ is the number of clients. As the size of the model or the number of clients increases, such overhead issue

---

[4]Note that the reveal in the DReLU protocol are masked intermediate values, rather than plain values.

(especially communication overhead) will become the main bottleneck for the scalability of the system. Currently, the system can only carry a small number of clients and medium-sized models. Hence, we view PrivHFL as a first step in constructing privacy-preserving protocols for HFL.

# 4 EVALUATION

## 4.1 EVALUATION SETUP

**Datasets and models.** We evaluate PrivHFL on three image datasets (SVHN, CIFAR10 and Tiny ImageNet). By default, we assume independent and identically distributed (IID) training data among clients, and the Non-IID setting can be found in Appendix B.3. For SVHN and CIFAR10, following CaPC we set the number of clients $n = 50$ and use VGG-7, ResNet-8 and ResNet-10 architectures as the clients' local models. For Tiny ImageNet, we use ResNet-14, ResNet-16, ResNet-18 architectures and set $n = 10$. Unless otherwise stated, we only report the accuracy after one iteration, and each model architecture is used by $n/3$ clients. Refer to Appendix F for more experimental setup.

**Cryptographic protocol.** We build PrivHFL on top of CryptGPU (Tan et al., 2021), but reimplement the underlying cryptographic protocols proposed in Section 3. We set the security parameter $\kappa$ as 128. As recommended by CryptGPU, we set secret-sharing protocols over the 64-bit ring $\mathbb{Z}_{2^{64}}$, and encode inputs using a fixed-point representation with 20-bit precision.

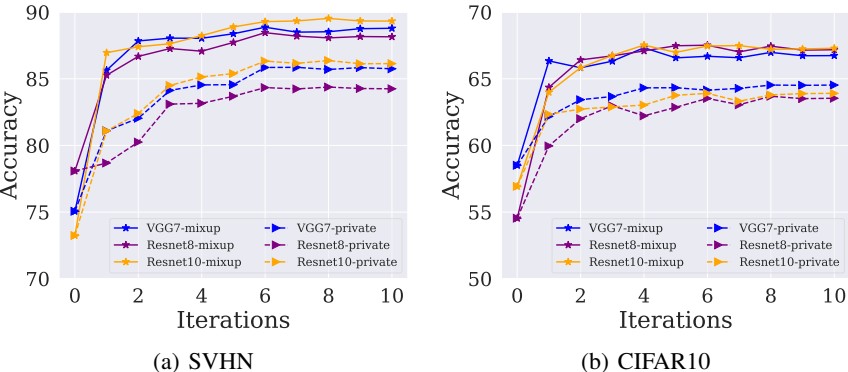

(a) SVHN               (b) CIFAR10

Figure 6: **The performance of PrivHFL on different query datasets: the private dataset and the synthetic datset based on mixup.** We report the average accuracy of each model architecture.

## 4.2 EVALUATION ON THE HETEROGENEOUS FEDERATED LEARNING

Table 1: **Test accuracy of PrivHFL on different fractions of participating clients and different numbers of query data.** We also report the model accuracy trained on the local dataset (i.e., *baseline* described in Section 3.1), and the model accuracy retrained on the private data-based query.

| | | SVHN | | | CIFAR10 | | | Tiny ImageNet | | |
| --- | --- | --- | --- | --- | --- | --- | --- | --- | --- | --- |
| | | $C = 0.6$ | $C = 0.8$ | $C = 1$ | $C = 0.6$ | $C = 0.8$ | $C = 1$ | $C = 0.6$ | $C = 0.8$ | $C = 1$ |
| **Baseline** | | | 75.46 | | | 56.66 | | | 22.26 | |
| **Private data** | | 79.43 | 79.56 | 80.29 | 60.82 | 61.01 | 61.49 | 24.89 | 25.11 | 25.23 |
| **mixup data** | 2.5K | 80.09 | 80.32 | 81.69 | 62.87 | 63.05 | 63.23 | 25.82 | 26.03 | 26.23 |
| | 5.0K | 83.32 | 83.52 | 83.82 | 63.04 | 63.44 | 63.69 | 26.22 | 26.46 | 26.75 |
| | 7.5K | 84.54 | 84.78 | 85.12 | 62.97 | 63.64 | 63.88 | 27.14 | 27.54 | 27.75 |
| | 10K | 84.58 | 84.97 | 85.62 | 63.79 | 63.82 | 64.56 | 27.67 | 28.19 | 28.46 |

Table 1 shows the detailed results on the performance improvement (i.e., accuracy gains) brought by PrivHFL, where we randomly select synthetic data to query instead of using active learning strategies. We observe that for SVHN and CIFAR10, the accuracy gain is about $10\%$ when we use $10K$

mixup samples. However, using the private dataset-based query strategy only increases the accuracy by about $4\%$. Table 1 also shows different accuracy gains as the number of mixup data increases, in which compared with $2.5K$ mixup data, we obtain $4.65\%, 1.33\%, 3.23\%$ increased accuracy using $10K$ samples on SVHN, CIFAR10 and Tiny ImageNet, respectively. This is because more synthetic data could provide a better coverage of natural dataset distributions. Besides, with the increase of participating fraction $C$, the accuracy improves slightly. Figure 6 and Figure 7 show the improvement of different heterogeneous models under different iterations and different numbers of query data. We observe that as the number of query data increases, PrivHFL consistently outperforms the baseline for the above datasets and heterogeneous models.

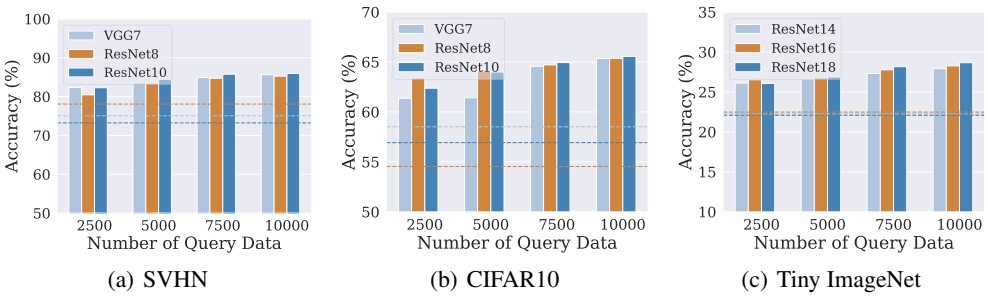

(a) SVHN      (b) CIFAR10      (c) Tiny ImageNet

Figure 7: **Test accuracy of heterogeneous models after PrivHFL as the number of query data increases.** Dashed lines represent the baseline, i.e., the test accuracy before executing PrivHFL.

**Ablation study of PrivHFL.** Figure 11(c) in Appendix B.1 illustrates the effectiveness of different data expansion methods. Figure 12 and Figure 16 in Appendix B.3 further illustrate the impact of active learning strategies and the number of private data samples on the test accuracy, respectively. Besides, we also show the test accuracy of PrivHFL on CIFAR10 and SVHN for different degrees of Non-IID-ness in Figure 15 in Appendix B.3.

## 4.3 EVALUATION ON THE CRYPTOGRAPHIC PROTOCOLS

Table 2: **Runtime (sec) of the three steps in PrivHFL's secure querying protocol.** The runtime of secure prediction represents the query time of each client on different models. CIFAR10 and SVHN have the same time due to the same input size and model architecture.

| Dataset | # Queries | 1. Query-data sharing | 2. Secure prediction | | | 3. Result aggregation |
|---|---|---|---|---|---|---|
| | | | VGG7 | ResNet8 | ResNet10 | |
| **CIFAR10 (SVHN)** | 1000 | 5.08 | 55.97 | 87.71 | 106.46 | 0.09 |
| | 2500 | 7.16 | 133.52 | 206.20 | 256.39 | 0.12 |
| | 5000 | 11.32 | 271.89 | 428.54 | 512.88 | 0.30 |
| | | | ResNet14 | ResNet16 | ResNet18 | |
| **Tiny ImageNet** | 1000 | 9.87 | 737.43 | 962.51 | 1075.13 | 0.18 |
| | 2500 | 18.78 | 1778.69 | 2356.33 | 2537.49 | 0.32 |

We mainly focus on the extra overhead caused by the secure querying phase. To clearly illustrate the efficiency of PrivHFL, unless otherwise specified, we only show the overhead of one communication round as shown in Section 3. Recall that our secure querying protocol consists of three phases, i.e., query-data sharing, secure prediction, and result aggregation. As shown in Table 2, the main cost of our framework comes from the second phase, where the evaluation of batched secure prediction is required. Specifically, it takes 4.5 minutes to evaluate 5000 query samples securely on VGG7 and CIFAR10. Besides, only 11.32 seconds and 0.3 seconds are spent on the query-data sharing and result aggregation phases. More runtime is required to evaluate Tiny ImageNet because of increased input size and model architecture.

To demonstrate the effectiveness of PrivHFL's secure prediction protocol, we compare with CrypT-Flow2 (Rathee et al., 2020) and CryptGPU (Tan et al., 2021). In Table 4 of Appendix B.2, we also

Table 3: **Runtime (sec) and communication cost (MB) of secure prediction for different methods on CIFAR10.** CrypTFlow2 (Rathee et al., 2020) is the SOTA 2-party protocol that contains two variants (OT-based and HE-based) and CryptGPU (Tan et al., 2021) is the SOTA 3-party GPU-friendly protocol. We report the cost of a single prediction on three models.

| Method | VGG7 | | ResNet8 | | ResNet10 | |
|---|---|---|---|---|---|---|
| | Time | Comm. | Time | Comm. | Time | Comm. |
| CrypTFlow2-OT (Rathee et al., 2020) | 39.22 | 15562.9 | 39.35 | 21261.6 | 55.76 | 28517.6 |
| CrypTFlow2-HE (Rathee et al., 2020) | 48.70 | 651.51 | 56.21 | 1110.39 | 97.46 | 1395.18 |
| CryptGPU (Tan et al., 2021) | 1.61 | 144.51 | 2.02 | 131.39 | 2.79 | 221.57 |
| This work (CPU) | 1.03 | 66.61 | 1.63 | 73.85 | 2.00 | 105.88 |
| **This work (GPU)** | **0.35** | **66.61** | **0.36** | **73.85** | **0.53** | **105.88** |

compare with CaPC (Choquette-Choo et al., 2021) that uses HE-Transformer as the building block. Table 3 summarizes this improvement for three models over CIFAR10. We observe that, PrivHFL achieves significant improvement on all the models in terms of runtime and communication costs. To be specific, PrivHFL requires 105.2-112.0 × less runtime and 233.6-269.3 × less communication compared with CrypTFlow2-OT. Similarly, compared with CrypTFlow2-HE, we show a greater advantage (i.e., 139.1-183.8 ×) in runtime, and a relatively small advantage (i.e., 9.7-15.0 ×) in communication overhead. In addition, we also test the performance under the CPU architecture. Notably, even comparing with CryptGPU, our weaker CPU setting obtains an improvement of roughly 1.5 × and 2 × in terms of computation and communication overheads, respectively.

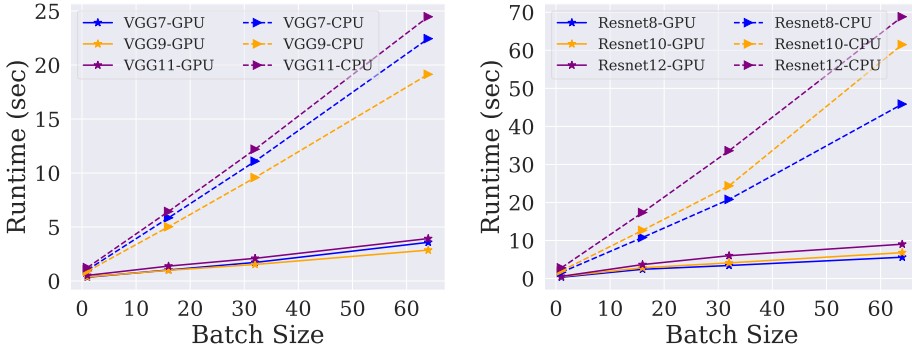

Figure 8: **Runtime (sec) of batch secure prediction on CPU and GPU settings as the batch size increases.** We report the result of VGG-style and ResNet-style networks on CIFAR10.

To further understand the impact of GPU acceleration on PrivHFL, we evaluate our secure prediction protocol under both the CPU and GPU settings with different batch size in Figure 8. We observe that the GPU-based secure prediction is always superior to the CPU analogs. As the batch size increases, the advantages of GPU-based protocols becomes more pronounced, e.g., the 9.5 × reduction on ResNet8 over a batch of 64 images.

## 5 CONCLUSION

In this paper, we propose a practical heterogeneous federated learning framework, which is independent on auxiliary datasets while provably guaranteeing the privacy of samples, model parameters and predictions. Extensive experiments demonstrate that PrivHFL outperforms prior art two orders of magnitude in efficiency and realizes about 10% accuracy gains. In the future, we will further improve the scalability of the system and integrate with advanced differential privacy mechanisms.

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

## A    HETEROGENEOUS FEDERATED LEARNING

In federated learning (FL), multiple clients collectively learn a single model with a fixed architecture, in which each client maintains a local model for the private dataset, while the server maintains a global model via aggregating the local model gradients from clients. Although successful, FL is often restrictive in practice, since it assumes all clients sharing the same model architecture. To tackle this problem, heterogeneous federated learning (HFL) is proposed, which enables clients to collaboratively train their own customized models that may differ in size or structure. Due to the heterogeneity in local models among clients, it's not possible to collaborate via sharing gradients like FL. As shown in Figure 9, generally, HFL uses a task-related auxiliary dataset to assist in collaborative learning. To be specific, each client first trains the local model with individual private dataset and then performs inference on the auxiliary dataset to obtain the prediction results, which are sent to the server to aggregate, rather than the local model gradients. Then, the server broadcasts the aggregated results to clients, and the clients will retrain local models based on the auxiliary dataset and the received predictions.

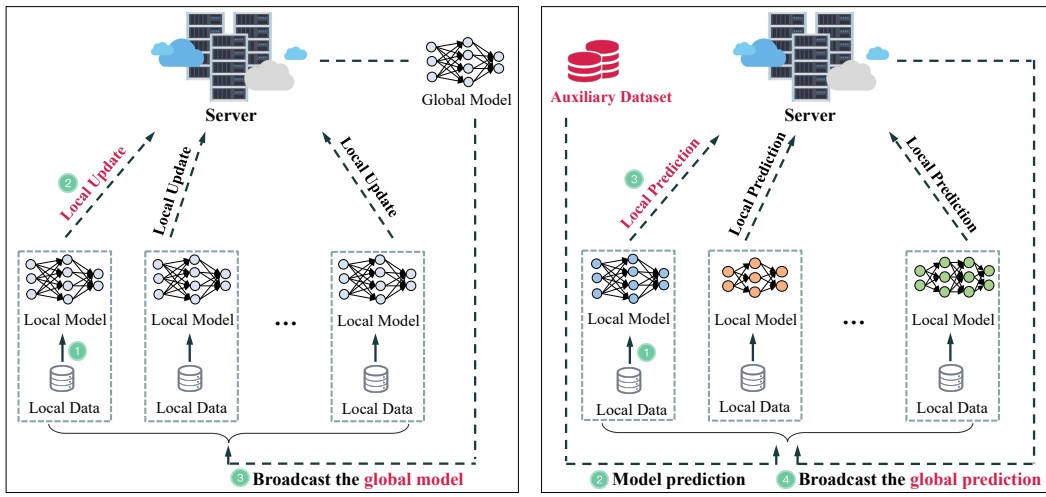

Figure 9: **Comparison with federated learning and heterogeneous federated learning**

## B    ADDITIONAL EXPERIMENTAL RESULTS

### B.1    UNDERSTANDING DATASET EXPANSION

Recall that we instantiate the dataset expansion method by leveraging mixup (Zhang et al., 2018) in Section 3.2. To explore the effectiveness of the mixup-based datset expansion method, we visualize the feature distribution of original data and synthetic samples on SVHN and CIFAR10. As shown in Figure 11(a) and Figure 11(b), the synthetic samples cover a larger part of the feature space and hence they should be more diverse and informative compared with original data. In other words, it could provide a good coverage of the manifold of natural samples. Therefore, learning the predictions from other clients on the synthetic samples can further improve the accuracy of local models.

Note that our dataset expansion method is universal and reconfigurable, which can be extended by leveraging other data augmentation strategies. Figure 11(c) gives the accuracy gains under various data augmentation strategies, including random sampling from Gaussian distribution, random flipping, cutmix and cutout. Cutmix (Yun et al., 2019) can be formulated as $\tilde{x}_{i,j} = M \cdot x_i + (1-M) \cdot x_j$, where $M \in \{0,1\}^{W \times H}$ is a binary mask matrix of size $W \times H$ to indicate the location of dropping out and filling from the two images $x_i$ and $x_j$. Cutout (DeVries & Taylor, 2017) augments the dataset with partially occluded versions of original samples. Figure 10 shows the overview of the results of these five strategies on an original image. As shown in Figure 11(c), the Gaussian

noise-based and random flipping-based dataset expansion strategies dramatically reduce the model accuracy. The main reason is that the two strategies do not necessarily model the problem domain distributions and do not provide informative natural images. In contrast, the strong data augmentation strategies, cutout and cutmix, are good choices for data generation in PrivHFL. Like mixup, these two methods can construct a big synthesized pool on a small private dataset that could provide a good coverage of the manifold of natural samples, to better distill knowledge of heterogeneous models.

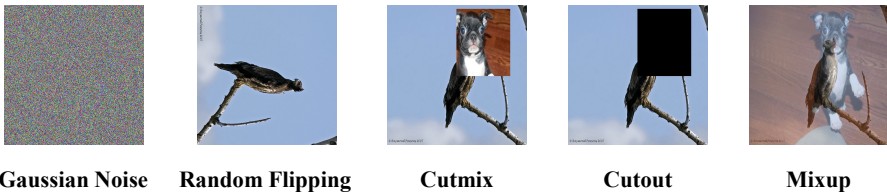

**Gaussian Noise**    **Random Flipping**    **Cutmix**    **Cutout**    **Mixup**

Figure 10: **Overview of the results of Gussian noise, random flipping, cutmix, cutout, and mixup data augmentation strategies.**

## B.2 EXPERIMENTAL RESULTS FOR SECURE PREDICTION

In the model querying phase, we leverage the GPU parallelism to process private querying on a batch of images, which effectively amortize the cost of private inference. In Table 4, we compare our GPU-friendly method on MNIST using the three model architectures (CryptoNets (Gilad-Bachrach et al., 2016), CryptoNets-ReLU (Gilad-Bachrach et al., 2016) and MLP (Boemer et al., 2019b)) with HE-Transformer, which also achieves batch-axis packing for private inference. Note that CaPC (Choquette-Choo et al., 2021) also adopt HE-Transformer as the building block. We can observe that PrivHFL is up to three orders of magnitude faster than HE-Transformer on the CryptoNets-ReLU and MLP models. The main reason is the efficiency of the protocol and the parallelism supported by the GPU. However, when evaluating CryptoNets that replaces ReLU activations with squared approximation, the improvement from PrivHFL will gradually decrease as the batch size increases, because the HE-based method does not require communication during the entire evaluation process. Note that approximations result in significant accuracy losses and degrades user experience, especially when evaluating modern large-scale models.

## B.3 EXPERIMENTAL RESULTS FOR HETEROGENEOUS FEDERATED LEARNING

**Impact of different active learning strategies.** Figure 12 depicts the impact of different active learning strategies, in which even random sampling can drastically improve the model accuracy by

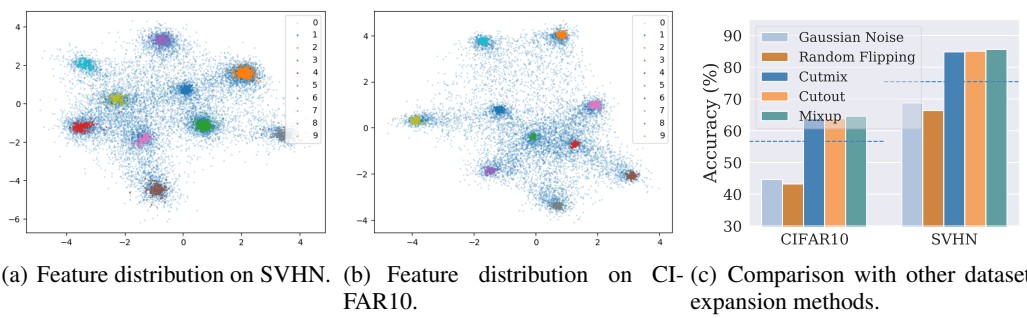

(a) Feature distribution on SVHN.    (b) Feature distribution on CIFAR10.    (c) Comparison with other dataset expansion methods.

Figure 11: **Understanding the importance of dataset expansion on improving model performance.** In (a) and (b), the bright colored points indicate the position of the original training data in the feature space, and the light blue points indicate the distribution of the expanded points based on mixup in the feature space. In (c), the left and right dashed lines are the baseline test accuracy on CIFAR10 and SVHN, respectively.

Table 4: **Runtime (sec) of secure prediction for HE-Transformer and our PrivHFL on MNIST as the batch size increases.** He-Transformer is the 2-party secure prediction protocol used in CaPC (Choquette-Choo et al., 2021).

| Model | BS=128 | | BS=256 | | BS=512 | | BS=1024 | | BS=2048 | |
|---|---|---|---|---|---|---|---|---|---|---|
| | Ours | HE-T | Ours | HE-T | Ours | HE-T | Ours | HE-T | Ours | HE-T |
| CNet1 | **0.07** | 17.75 | **0.14** | 17.56 | **0.25** | 17.62 | **0.49** | 17.77 | **0.99** | 17.67 |
| CNet2 | **0.07** | 48.83 | **0.15** | 70.14 | **0.26** | 112.42 | **0.51** | 201.42 | **1.01** | 369.51 |
| MLP | **0.05** | 65.01 | **0.06** | 86.37 | **0.10** | 129.81 | **0.16** | 216.61 | **0.29** | 391.13 |

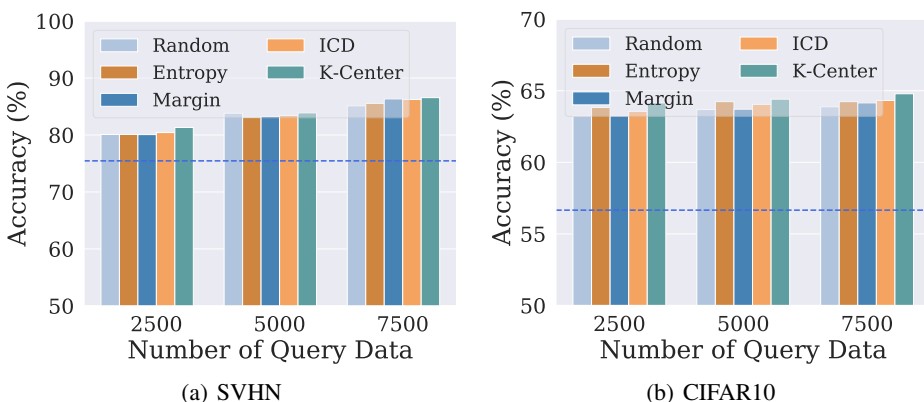

(a) SVHN          (b) CIFAR10

Figure 12: **Using PrivHFL with active learning strategies to improve model accracy on SVHN and CIFAR10 datasets.** Dashed line represents the baseline accuracy of models, and the histogram represents the accuracy of the model after PrivHFL based on different active learning strategies.

4.46%-9.66% and 6.57%-7.22% on SVHN and CIFAR10 datasets. Nonetheless, we can also achieve additional benefits leveraging active learning, such as 1.44% gains on SVHN with the entropy sampling and 0.8 % gains on CIFAR10 with the margin sampling.

**Impact of the Non-IID-ness degree.** We use the Dirichlet distribution $Dir(\alpha)$ as in Lin et al. (2020) to simulate disjoint Non-IID training data. Figure 14 visualizes the distributions of Non-IID samples among clients with different $Dir(\alpha)$ on CIFAR10 dataset, where the number of clients $n = 20$. The value of $\alpha$ controls the degree of Non-IID-ness, in which a smaller $\alpha$ indicates higher degree of Non-IID-ness. When $\alpha = 100$, the distribution closes to uniform sampling. When $\alpha = 0.5$, the distribution of samples of each class among clients is extremely uneven. Figure 15 shows the test accuracy of PrivHFL on SVHN and CIFAR10 for different degrees of Non-IID-ness. We can observe that the higher the degree of Non-IID-ness, the lower the accuracy of the model. However, in this case, the mixup-based data expansion method can still significantly improve the performance of models.

**Impact of the private data volume.** Figure 16 illustrates the impact of private data volume on the test accuracy. We can observe that as the private data volume increases, the performance of models is on the rise. The main reason is that the model can learn more knowledge from more private training data, and it can also generate more mixup synthetic samples to query, so as to promote the sharing of model knowledge.

**Impact of the normalization operation.** As described in Figure 5, we directly aggregate the predicted logits instead of the normalized values. There are two major reasons: i) The normalization operation in the secure computation will introduce extra communication and computation overheads, since it contains costly multiplication and division operations (Knott et al., 2021). ii) The diverse logits of the heterogeneous models may contain informative content and hence facilitate model accuracy. This phenomenon is particularly prominent on complex datasets. As shown in Figure 13,

Table 5: **Impact of the coefficient $\lambda$ in mixup on CIFAR10.**

| mixup $\lambda$ | 0.1 | 0.2 | 0.3 | 0.4 | 0.5 | 0.6 | 0.7 | 0.8 | 0.9 |
|---|---|---|---|---|---|---|---|---|---|
| **Accuracy** | 61.93 | 61.82 | 61.42 | 61.37 | 61.38 | 61.35 | 61.80 | 61.53 | 61.73 |

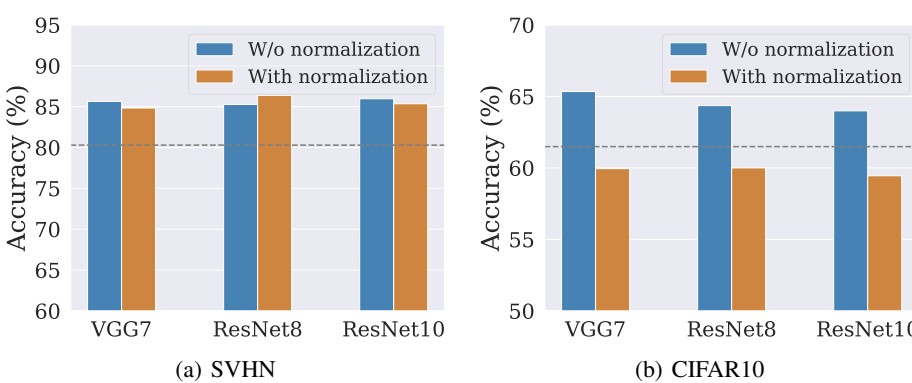

(a) SVHN  (b) CIFAR10

Figure 13: **Impact of the normalization operation on the accuracy of heterogeneous models on SVHN and CIFAR10 datasets.** Dashed line represents the average accuracy of heterogeneous models when only private data is used for query.

the model accuracy on CIFAR10 without normalization is about 5% higher than that with normalization. On SVHN, the influence of normalization is negligible, since this classification task is relatively simple.

**Impact of the coefficient $\lambda$ in mixup.** Table 5 shows the impact of different mixup coefficients $\lambda$ on the model accuracy over CIFAR10. For each result, we only use one coefficient to generate synthetic images. We can observe that in different coefficient values, the model accuracy remains almost unchanged, and hence the coefficient of mixup has a negligible effect on the accuracy of the heterogeneous models in PrivHFL. Therefore, we set $\lambda \in [0.1, 0.9]$ with an interval of 0.1 in our experiments to generate more diverse synthetic images.

## C  MISSING DETAILS ON PRIVHFL

### C.1  ALGORITHMIC DESCRIPTION

Algorithm 1 gives the detailed description of the PrivHFL framework. In PrivHFL, each client first trains the local model $M_j$ to convergence on the private dataset. Later, clients improve the performance of local models based on the knowledge of others via the predictions on the synthetic dataset.

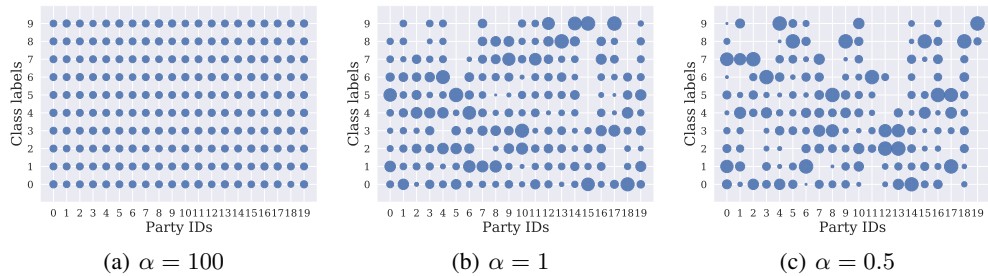

(a) $\alpha = 100$  (b) $\alpha = 1$  (c) $\alpha = 0.5$

Figure 14: **Visualization of Non-IID-ness among clients with different Dirichlet distribution $\alpha$ values on CIFAR10 dataset.** The size of scattered points indicates the number of training samples for a class available to that client.

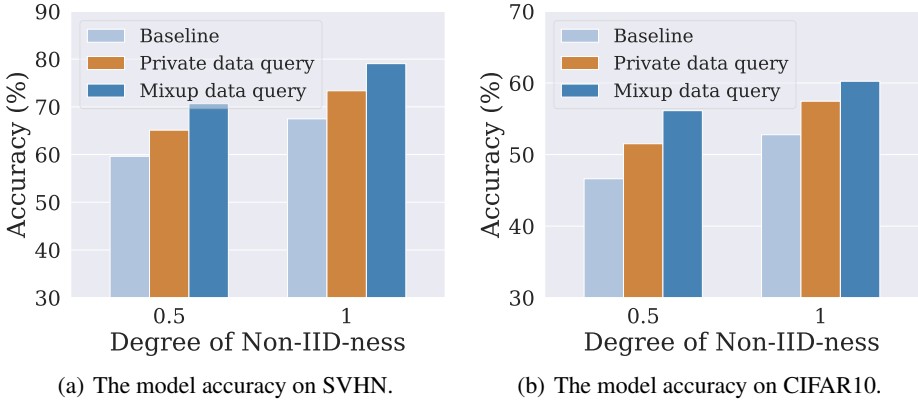

(a) The model accuracy on SVHN.

(b) The model accuracy on CIFAR10.

Figure 15: **The test accuracy of PrivHFL on SVHN and CIFAR10 with different degrees of Non-IID-ness.** The number of clients $n = 20$.

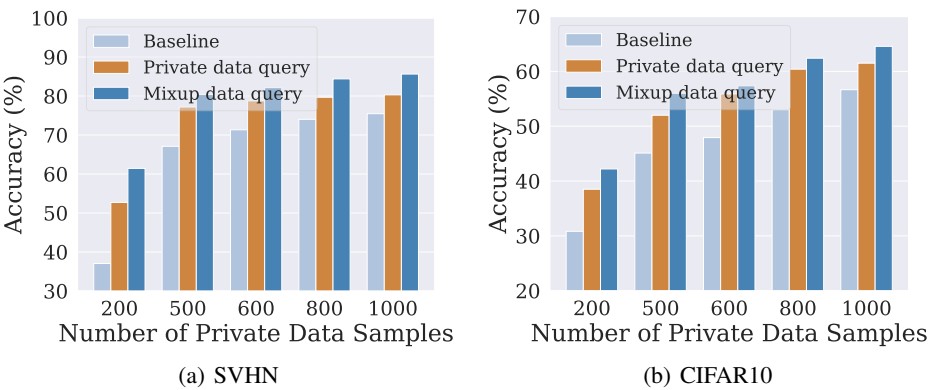

(a) SVHN

(b) CIFAR10

Figure 16: **Impact of different amounts of private data on the accuracy of models on SVHN and CIFAR10.** The number of clients $n = 50$, and the number of private data samples for each client ranges from 200 to 1000.

Therefore, they need to first construct a big unlabeled data pool based on the small number of private data, and then actively choose a subset from the pool as the query data in each iteration. Next, they perform the secure querying protocol based on the query data, and obtains the final predictions, which will be used to retrain the local model. Query-data generation and secure querying process (refer to Section 3 for more details) will iterate $iter$ times until the model $M_j$ achieves pre-defined performance.

---

**Algorithm 1** The PrivHFL framework.

---

**Input:** Each client $P_j$, $j \in [n]$, holds a private dataset $\mathcal{D}_j$ and a customized local model $M_j$. $iter$ denotes the number of training iterations, where in HFL an iteration means that all clients have completed one knowledge transfer. $B$ denotes the number of the query dataset and $\mathcal{C}$ denotes the set of selected answering parties in current query phase.

**Output:** Trained models $M_j$, $j \in [n]$.

1: **for** each $j \in [n]$ **do**
2:    $P_j$ locally trains the local model $M_j$ on $\mathcal{D}_j$ using the stochastic gradient descent optimization.
3: **end for**
4: **for** each $iter$ **do**
5:    **for** each querying party $P_Q^j$, $j \in [n]$ **do**
6:       $P_Q^j$ generates an unlabeled synthetic pool $\mathcal{D}_{pool}^j$ with its own private dataset $\mathcal{D}_j$ by utilizing the dataset expansion method in Section 3.2.
7:       $P_Q^j$ applies the active learning methods in Section C.3 to select the query dataset $\{x_b\}_{b \in [B]}$ from $\mathcal{D}_{pool}^j$.
8:       **for** each answering party $P_A^i$, $i \in \mathcal{C}$ **do**
9:          $P_Q^j$ secret-shares $\{[x_b]\}_{b \in [B]}$ with $P_A^i$ and the server, based on the protocol $\Pi_{\text{Share}}$.
10:         $P_A^i$, $P_Q^j$ and the server jointly perform the secure querying protocol in Section 3.3.
11:         $P_A^i$ secret-shares the predictions $\{[y_b^i]\}_{b \in [B]}$ to $P_Q$ and the server.
12:       **end for**
13:       $P_Q^j$ computes $\{y_b\}_{b \in [B]}$ with $y_b = \sum_{i \in \mathcal{C}} y_b^i$ via performing the secure result aggregation protocol $\Pi_{Agg}$ with the server.
14:       $P_Q^j$ retrains its local model based on the query dataset $\{x_b, y_b\}_{b \in [B]}$ and private dataset $\mathcal{D}_j$.
15:    **end for**
16: **end for**

---

## C.2   Graphic depiction of end-to-end secure prediction

Figure 17 gives a graphic depiction to illustrate the end-to-end secure prediction implemented across all the layers, where the inputs are secret-shares of image $x$, i.e., $[x]_0$ and $[x]_1$ (as shown in Figure 20). The inputs first pass through a convolutional layer that mainly contains matrix multiplication operations $\omega_1 \cdot x$ ($\omega_1$ is the parameter of this layer) and can be implemented by the protocol $\Pi_{\text{Matmul}}$ in Figure 3. The outputs of this layer are in the secret sharing form, i.e., $[y_1]_0$ and $[y_1]_1$ obtained by $P_A$ and the server, respectively. For the ReLU layer, recall that $\text{ReLU}(y_1) = y_1 \cdot \text{DReLU}(y_1)$. Therefore, the protocol $\Pi_{\text{DReLU}}$ in Figure 4 is executed first to obtain $\text{DReLU}(y_1)$'s shares $[\text{DReLU}(y_1)]_0$ and $[\text{DReLU}(y_1)]_1$. Then, $P_A$ and the server invoke an instance of the protocol $\Pi_{\text{Matmul}}$, and obtain $[y_2]_0$ and $[y_2]_1$, respectively. The outputs of $\Pi_{\text{DReLU}}$ are the inputs of the subsequent MaxPooling layer. As described in Section 3.3 (iii), MaxPooling on $n$ values can be converted into $n - 1$ ReLU operations. Therefore, the output of this layer is also in the secret sharing form. When the inference goes to the final fully-connected layer with inputs $[y_{n-1}]_0$ and $[y_{n-1}]_1$ owned by $P_A$ and the server, respectively, the protocol $\Pi_{\text{Matmul}}$ is executed. In the end, $P_A$ and the server obtain the secret-shares of the predicted logit, i.e., $[logit]_0$ and $[logit]_1$, respectively.

## C.3   Active learning strategies

In order to choose query data that most likely contribute to improve the performance of local models, inspired by CaPC (Choquette-Choo et al., 2021), active learning is adopted. To be specific, active learning allows local models to actively choose the data from which they learn, which con-

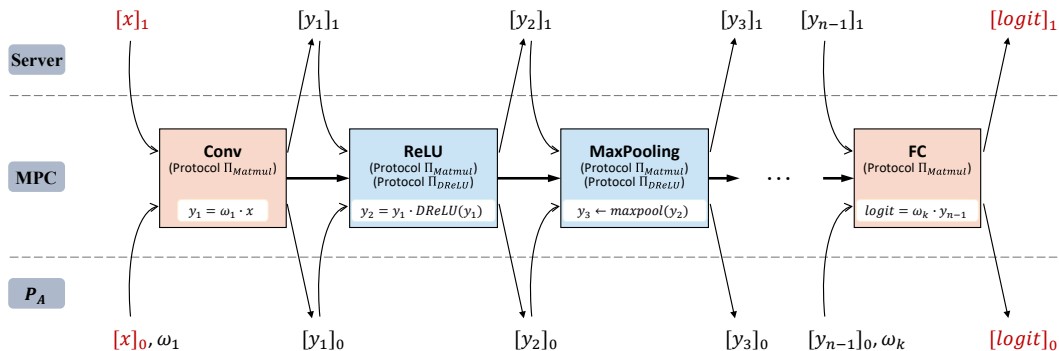

Figure 17: **The whole private inference implemented across all the layers.** Orange boxes represent linear layers (including convolutional/fully-connected/AvgPooling layers), and blue boxes represent non-linear layers (including ReLU/MaxPooling layers).

tains various pool sampling strategies to estimate the informativeness and diversity of unlabeled samplings. These strategies can be classified into two categories. One important class is that of uncertainty-based approaches, which try to find hard samples using heuristics like highest entropy, such as margin sampling and entropy sampling. There are recent optimization-based methods, such as greedy-k-center sampling to obtain a diverse subset of hard samples.

**Definition 1 (Margin Sampling (Scheffer et al., 2001)).** Margin sampling assumes that the most informative samples are those which fall within this margin of decision boundary. Formally, given an unlabeled dataset $D$, it can be represented as:

$$\hat{\mathbf{x}} = \arg\min_{x \in D} \left\{ \min_{\omega} |f(x, \omega)| \right\} \tag{2}$$

where $f(x, \omega)$ represents the distance of the data sample $x$ from the hyperplane for class $\omega$.

**Definition 2 (Entropy Sampling (Shannon, 2001)).** Using entropy as an uncertainty measure as follows:

$$\hat{\mathbf{x}} = \arg\max_{x \in D} - \sum_i P_\theta(y_i \mid x) \log P_\theta(y_i \mid x) \tag{3}$$

where $P_\theta(y \mid x)$ is the conditional label distribution of the model, $y_i$ ranges over all possible labels.

**Definition 3 (Greedy-k-center Sampling (Sener & Savarese, 2018)).** Solving the $k$-center problem, i.e., choosing $b$ center points such that the largest distance between a data point and its nearest center is minimized. Formally, this goal is defined as:

$$\min_{\mathbf{s}^1:|\mathbf{s}^1| \leq b} \max_i \min_{j \in \mathbf{s}^1 \cup \mathbf{s}^0} \Delta(\mathbf{x}_i, \mathbf{x}_j) \tag{4}$$

where $\mathbf{s}^1$ is new chosen center points, $\mathbf{s}^0$ is the current training set.

Besides, to better balance uncertainty and diversity, we introduce informative-and-diverse sampling called *ICD*, which chooses query data based on informative and diverse criteria using margin and cluster-based sampling methods. It will return highest uncertainty lowest margin points while maintaining same distribution over clusters as entire dataset.

## C.4 MORE BACKGROUND ON CRYPTOGRAPHY

### C.4.1 SECRET SHARING AND MULTIPLICATION TRIPLES

As shown in Section 2.3, in PrivHFL, we utilize additive secret sharing to protect the privacy of sensitive information. Assume there are two secret-shared values $[x]$ and $[y]$ that owned by two parties, addition and subtraction operations ($[z] = [x] \pm [y]$) can be done locally without any communication, which is realized as $[z]_i = [x]_i \pm [y]_i \mod 2^{64}$ by each party $P_i$, $i \in \{0, 1\}$.

$$z = xy = ([x]_0 + [x]_1)([y]_0 + [y]_1) = \overbrace{[x]_0[y]_0}^{P_0} + \overbrace{[x]_1[y]_1}^{P_1} + [x]_0[y]_1 + [x]_1[y]_0 \tag{5}$$

Multiplying two secrets, i.e., $[z] = [x] \cdot [y]$, however, is evaluated by using Beaver Multiplication Triple (Demmler et al., 2015). The triple refers to $(a, b, c)$ with the constraint $c = ab$, which can be generated using cryptographic techniques (Demmler et al., 2015) or a trusted dealer (Riazi et al., 2018). To be specific, as shown in Eq.5, $[x]_i[y]_i$ can be computed by party $P_i$ locally, but $[x]_i[y]_{1-i}$ evaluated as follows. Taking $[x]_0[y]_1$ as an example, we suppose $P_0$ and $P_1$ already hold the triples $(a, [c]_0)$ and $(b, [c]_1)$, respectively. $P_0$ first sends $[x]_0 + a \pmod{2^{64}}$ to $P_1$, while $P_1$ sends $[y]_1 - b \pmod{2^{64}}$ to $P_0$. Then $P_0$ computes one share of $[x]_0[y]_1$ as $[x]_0([y]_1 - b) - [c]_0 \pmod{2^{64}}$, and $P_0$ computes another as $([x]_0 + a)[y]_1 - [c]_1 \pmod{2^{64}}$, locally. In this way, both parties can achieve secure multiplication operation, and the outputs are still in the form of secret sharing.

### C.4.2 DIFFIE-HELLMAN KEY AGREEMENT PROTOCOL

In order to allow two parties to agree the same secret key on an insecure channel, the Diffie-Hellman Key Agreement (DH) protocol (Diffie & Hellman, 1976) was proposed. In PrivHFL, the DH protocol is used to generate the consistent PRG seeds between clients, which consists of the following three steps:

- DH.param$(k) \rightarrow (\mathbb{G}, g, q, H)$ generates a group $\mathbb{G}$ of prime order $q$, along with a generator $g$, and a hash function $H$.

- DH.gen$(\mathbb{G}, g, q, H) \rightarrow (x_i, g^{x_i})$ randomly samples $x_i \in \mathbb{Z}_q$ as the secret key and let $g^{x_i}$ as the public key.

- DH.agree$(x_i, g^{x_j}, H) \rightarrow s_{i,j}$ outputs the *seed* $s_{i,j} = H((g^{x_j})^{x_i})$.

In the DH protocol, correctness requires that for any key pairs $(x_i, g^{x_i})$ and $(x_j, g^{x_j})$ generated by two paries $P_i$ and $P_j$ using DH.gen under the same parameters $(\mathbb{G}, g, q, H)$, DH.agree$(x_i, g^{x_j}, H) = $ DH.agree$(x_j, g^{x_i}, H)$. Besides, in the honest-but-curious adversary setting, security requires that for any adversary who steals $g^{x_i}$ and $g^{x_j}$ (but neither of the corresponding $x_i$ and $x_j$), the agreed secret $s_{i,j}$ derived from those keys is indistinguishable from a uniformly random value (Abdalla et al., 2001).

**Seed generation between clients without directly communication.** The communication improvement of our PrivHFL is largely derived from the application of PRGs. These allow any two clients to jointly generate same (pseudo-) random values that are used in MPC protocols without communication. It is trivial to construct PRG seeds $Sk_{SA}$ (between $P_A$ and the server) and $Sk_{SQ}$ (between $P_Q$ and the server). For instance, the server first generates $Sk_{SA}$ and $Sk_{SQ}$, and then sends them to $P_A$ and $P_Q$, respectively. However, this is challenging for constructing $Sk_{QA}$, given that direct communication channels can not be constructed between clients. To tackle this issue, we generate seed $Sk_{QA}$ using the DH protocol (Diffie & Hellman, 1976), a classic algorithm for exchanging secret keys securely. Specifically, we first call the above DH.gen algorithm, and communicate the resulting public keys of two clients via the server as an intermediary. After that, the same seed is locally generated by the two clients via the DH.agree algorithm. We show the protocol $\Pi_{\text{Seed}}$ in Figure 18 and its security follows from the security of the DH protocol (Abdalla et al., 2001).

Figure 18: Secure PRG seed generation protocol $\Pi_{\text{Seed}}$

## C.5 EXTENDING CAPC TO COMMUNICATION-LIMITED SETTINGS

By carefully designing protocols, CaPC (Choquette-Choo et al., 2021) can also be extended to a setting where there is no direct communication between clients. However, as analyzed below, such extension comes at the cost of increased communication overhead. Therefore, our PrivHFL has greater advantages compared to the following modified protocols.

Recall that HE-Transformer (Boemer et al., 2019b;a) provides two secure prediction schemes, i.e., a pure HE-based scheme and a hybrid scheme combining HE and GC. Given that the HE-Transformer framework is employed in CaPC, we design two customized protocols for HE-Transformer to achieve such communication requirement. Although it is trivial to extend CaPC equipped with the HE-based scheme to the above communication-limited settings, it has two key problems: 1) activation functions need to be approximated as low-degree polynomials, which leads to serious accuracy loss; 2) due to the inherent high computational complexity, HE-based secure prediction is difficult to extend to large-scale models. For completeness, we briefly describe the extension procedure. $P_Q$ first encrypts the query samples and asks the server to pass them to $P_A$. After that, $P_A$ evaluates secure prediction non-interactively in the ciphertext environment. Finally, $P_A$ sends encrypted masked predictions (due to privacy-preserving aggregation in CaPC) back to $P_Q$ via the server acting as a intermediary. As mentioned above, the HE-based method is impractical and below, we elaborate on the extension of CaPC that uses the hybrid scheme as a building block. In CaPC, secure predictions are executed between $P_Q$ and $P_A$. To tackle the communication limitation, we can employ secure predictions between the server and $P_A$. We discuss the modified algorithms of the linear layer and the non-linear layer separately. In the linear layer, 1) $P_Q$ encrypts query samples with HE and sends the ciphertext to $P_A$ through the server[5]. 2) $P_A$ evaluates linear layers locally, such as convolution and full-connected layers, and returns the encrypted masked results to $P_Q$ through the server. 3) $P_Q$ decrypts to obtain the masked results and sends it to the server. As a result, the results of linear layers are shared between the server and $P_A$. For the non-linear layer, given that the server and $P_A$ hold shares of the linear layer's results, the two parties (rather than $P_A$ and $P_Q$ in CaPC) call the GC protocol to evaluate the nonlinear layer function.

Next, we analyze the efficiency of the modified hybrid protocols. For the computational cost, the modified secure prediction protocol is exactly the same as that of CaPC, and thus we mainly focus on the communication cost. In the linear layer, the modified protocol adds the communication overhead of a ciphertext (two ciphertexts for the input layer) and a plaintext of the result of the linear layer. For the non-linear layer, the communication overhead is not increased, but the overhead between $P_A$ and $P_Q$ is transferred to the server and $P_A$. In summary, although CaPC can be extended to scenarios with limited communication, they sacrifice the efficiency of secure predictions. Therefore, PrivHFL shows better adaptability and efficiency in scenarios where there is no direct communication between clients.

## C.6 GPU ACCELERATION

Existing general-purpose platforms on the GPUs such as NVIDIA's CUDA are designed to operate on floating-point inputs (Tan et al., 2021). On the contrary, the shares in our protocol is embedded in the ring $\mathbb{Z}_{2^{64}}$. Thus, we need to convert the ring operations into 64-bit floating point operations to obtain GPU support. Specifically, a 64-bit floating-point number has 1 bit *sign*, 11 bits *exponent*, and 52 bits *precision*, which can exactly represent all integers in the interval $\left[-2^{52}, 2^{52}\right]$. Therefore, the multiplication $E \cdot F$ can be correctly computed and recovered over the integers if and only if $E, F \in \mathbb{Z} \cap \left[-2^{26}, 2^{26}\right]$. Based on this observation, we decompose each input $E$ and $F$ in $\mathbb{Z}_{2^{64}}$ into 4 blocks, and the values in each block are represented by a 16-bit value. Thus, the integer multiplication is converted to 16 (exactly, 10) floating-point operations. Specifically, $E$ is rewritten as $E_0 + 2^{16}E_1 + 2^{32}E_2 + 2^{48}E_3$, and $F = F_0 + 2^{16}F_1 + 2^{32}F_2 + 2^{48}F_3$. The multiplication can be represented as follows:

$$
\begin{aligned}
E \cdot F &= (E_0 + 2^{16}E_1 + 2^{32}E_2 + 2^{48}E_3) \cdot (F_0 + 2^{16}F_1 + 2^{32}F_2 + 2^{48}F_3) \\
&= E_0F_0 + 2^{16}E_1F_0 + 2^{32}E_2F_0 + 2^{48}E_3F_0 + 2^{16}F_1E_0 + 2^{32}F_1E_1 + 2^{48}F_1E_2 \\
&\quad + 2^{32}F_2E_0 + 2^{48}F_2E_1 + 2^{48}F_3E_0
\end{aligned} \tag{6}
$$

---

[5]To be more precise, this step is for the input layer. In the hidden layer, one of the input shares of the linear layer should be encrypted by the server and sent to $P_A$.

 After that, computing $E \cdot F$ from the pairwise products requires element-wise additions and scalar multiplications, which can be executed by optimized CUDA kernels on 64-bit integer values.

# D    RELATED WORK

Table 6: **Comparison with prior works on properties necessary for federated learning**

| Framework | Privacy | | Usability | | Efficiency | |
|---|---|---|---|---|---|---|
| | Data Privacy | Model Privacy | Model Heterogeneity | w/o Dataset Dependency | GPU Compatibility | Protocol Efficiency |
| Bonawitz et al. (2017) | ✓ | ✗ | ✗ | ✓ | ✗ | ✓ |
| Bell et al. (2020) | ✓ | ✗ | ✗ | ✓ | ✗ | ✓ |
| Sav et al. (2021) | ✓ | ✓ | ✗ | ✓ | ✗ | ✗ |
| Jayaraman & Wang (2018) | ✓ | ✗ | ✗ | ✓ | ✗ | ✓ |
| Li & Wang (2019) | ✗ | ✓ | ✓ | ✗ | ✓ | - |
| Choquette-Choo et al. (2021) | ✓ | ✓ | ✓ | ✗ | ✗ | ✗ |
| Lin et al. (2020) | ✗ | ✗ | ✓ | ✗ | ✓ | - |
| Sun & Lyu (2021) | ✗ | ✓ | ✓ | ✗ | ✓ | ✓ |
| Diao et al. (2021) | ✗ | ✗ | ✓ | ✓ | ✓ | - |
| This work | ✓ | ✓ | ✓ | ✓ | ✓ | ✓ |

## D.1    HETEROGENEOUS FEDERATED LEARNING

Federated learning achieves collaboration among clients via sharing model gradients. While successful, it still faces many challenges, among which, of particular importance is the heterogeneity that appear in all aspects of the learning process. This consists of system heterogeneity (Diao et al., 2021), model heterogeneity (Li & Wang, 2019) and statistical heterogeneity (Zhu et al., 2021). Statistical heterogeneity means that clients' data from real-world comes from distinct distributions (i.e., Non-IID data), which may induce deflected local optimum. Solving the statistical heterogeneity has been extensively studied, such as Dinh et al. (2020); Zhu et al. (2021); Yurochkin et al. (2019); Fallah et al. (2020); Yoon et al. (2021)[6], and is out-of-the-scope of this work. However, our PrivHFL may help alleviate the statistical heterogeneity due to customized model design and knowledge distillation-based aggregation rule.

Our work mainly focuses on model heterogeneity that has been explored in recent works (Li & Wang, 2019; Lin et al., 2020; Choquette-Choo et al., 2021), while the issue of system heterogeneous is alleviated through the resource-customized model architecture design. In particular, Li & Wang (2019) proposed the first federated learning framework FedMD supporting heterogeneous models by combining transfer learning and knowledge distillation techniques. They first used a public dataset to pre-train the model and transferred to the task of private dataset. After that, to exchange the knowledge, each client used the public data and the aggregated predictions from others as carrier for knowledge distillation. To further improve test accuracy, Lin et al. (Lin et al., 2020) proposed FedDF, similar to FedMD, which also used model distillation technique for knowledge sharing. The difference is that they first performed FedAvg on clients' local models and integrated knowledge distillation on the aggregated model. The dependence on model averaging leads to limited model heterogeneity. Besides, Diao et al. (Diao et al., 2021) focused on heterogeneous clients equipped with different computation and communication capabilities. In their framework, each client only updated a subset of global model parameters through varying the width of hidden channels, which reduces the computation and communication complexity of local models. However, this approach only learns a single global model, rather than unique models designed by clients. Moreover, the above three methods rarely consider the issue of privacy leakage from the prediction results.

The privacy protection techniques (i.e., secure aggregation) have been studied in federated learning (Bonawitz et al., 2017; Bell et al., 2020; Sav et al., 2021; Jayaraman & Wang, 2018). However, these techniques can not be directly extended to privacy-preserving HFL. More recently, Sun & Lyu

---

[6]This work also uses mixup in federated learning, but aims to address the challenge of non-iid data.

(2021) proposed a noise-free differential privacy solution for heterogeneous federated learning to guarantee each client's privacy. However, as shown in Jayaraman & Evans (2019), there is a huge gap between the upper bounds on privacy loss analyzed by advanced mechanisms and the effective privacy loss. Thus, differentially private mechanisms offer undesirable utility-privacy trade-offs. To further formally guarantee the privacy, Choquette-Choo et al. (2021) leveraged secure multi-party computation (MPC), homomorphic encryption (HE) and privately aggregated teacher models techniques to realize confidential and private collaborative learning. Specifically, clients learn from each other collaboratively utilizing a secure inference strategy based on MPC and HE protocols and a private aggregation method. As noted in the Introduction, CaPC's use of heavy cryptography leads to significant efficiency and communication overheads. Besides, existing methods require an auxiliary dataset to implement heterogeneous federated learning. However, data collection could be unrealistic in many real-world scenarios due to various reasons, such as privacy concerns and rare classes. Therefore, our work is dedicated to solving two key challenges. The first is to relax the assumption of relying on public datasets, and the second is to design an efficient cryptographic protocol for knowledge transfer during the model aggregation.

## D.2 PRIVATE NEURAL NETWORK PREDICTION

Neural networks present a challenge to cryptographic protocols due to their unique structure and exploitative combination of linear computations and non-linear activation functions. In real scenarios, model inference can be viewed as a two-party computation case, where one party with private data wants to obtain prediction results from the other party who owns the model. During the whole process, the cryptographic protocols, typically HE and MPC, are applied to ensure the confidentiality of the private data and model.

Existing works (Boemer et al., 2019b; Gilad-Bachrach et al., 2016; Brutzkus et al., 2019) support pure HE protocols for secure predictions. Typically, nGraph-HE (Boemer et al., 2019b;a) allows linear computations using CKKS homomorphic encryption scheme. However, since a solution that builds upon HE protocols should be restricted to compute low degree polynomials, the non-polynomial activation functions, such as MaxPooling and ReLU, are forced to be evaluated in the clear by the party who owns private query data. This leaks the feature maps, from which adversaries may deduce the model weights. To solve this problem, Gilad-Bachrach et al. (2016) and Chen et al. (2019) use low-degree polynomial approximation to estimate non-linear functions. Unfortunately, it will affect the accuracy of predictions, while leading to huge computation overhead.

On the other hand, server libraries (Mohassel & Zhang, 2017; Knott et al., 2021; Wagh et al., 2019; Shen et al., 2020) employ primarily MPC technology in secure predictions, which provides support for linear and non-linear activations through the use of oblivious transfer (OT), garbled circuit (GC) and secret sharing. For example, CryptTen (Knott et al., 2021) performs linear operations based on $n$-out-of-$n$ additive secret sharing over the ring, $\mathbb{Z}_{2^{64}}$. However, it uses boolean secret sharing for the non-linear operations, which will result in a higher communication round. CrpytGPU (Tan et al., 2021) builds on CrypTen, working in a 3-party setting using *replicated secret shares*. Although the scalability is poor, it introduces an interface to losslessly embed cryptographic operations over secret-shared values in a discrete somain into floating-point operations, which can implement the whole inference process on the GPU.

Many other works focus on hybrid protocols, in which they combines the advantages of HE and MPC to improve prediction efficiency (Juvekar et al., 2018; Mishra et al., 2020; Rathee et al., 2020). CrypTFlow2 (Rathee et al., 2020) points out that currently, it was not clear whether HE-based linear operations would provide the best latency. Therefore, the authors implement two class of protocols, HE-based and OT-based, for linear operations. For non-linear layers, they also design efficient protocols based on OT. It turns out that in a WAN setting, HE-based inference is always faster and in a LAN setting OT and HE are incomparable. HE-transformer employs nGraph-HE for evaluation of linear operations, and ABY framework (Demmler et al., 2015) for GC to evaluate non-linear functions. Since non-linear operations cannot be parallelized between query data, GC is inefficient, especially for large networks with thousands of parameters. In contrast, our PrivHFL avoids the use of heavy cryptographic tools, and only employs additive secret sharing to achieve high efficiency, confidentiality and practicability.

# E    SECURITY ANALYSIS

Our security proof follows the standard ideal-world/real-world paradigm (Canetti, 2001): in real-world, three parties interact according to the protocol specification, and in ideal-world, they have access to a ideal functionality. When a protocol invokes another sub-protocol, we use $\mathcal{F}$-hybrid model for security proof by replacing the sub-protocol with the corresponding functionality. Note that our proof works in the $\mathcal{F}_{PRG}$-hybrid model where $\mathcal{F}_{PRG}$ represents the ideal functionality corresponding to the protocol PRG. The executions in both worlds are coordinated by the environment Env, who chooses the inputs to parties and plays the role of a distinguisher between the real and ideal executions. We will show that the real-world distribution is computationally indistinguishable to the ideal-world distribution.

**Theorem 1.** $\Pi_{\text{Share}}$ securely realizes the functionality $\mathcal{F}_{\text{Share}}$ in the $\mathcal{F}_{\text{PRG}}$-hybrid model.

**Proof.** Note that $P_Q$ and $P_A$ receive no messages in the protocol, and hence the sharing protocol is trivially secure against corruption of $P_Q$ and $P_A$. Next, the only message that the server receives is the value $[x]_1$. However, $[x]_1 = x - r$, where given the security of PRG, $r$ is a random value unknown to the server. Thus, the distribution of $[x]_1$ is uniformly random from the server's view and the information learned by the server can be perfectly simulated. $\qquad\square$

**Theorem 2.** $\Pi_{\text{Matmul}}$ securely realizes the functionality $\mathcal{F}_{\text{Matmul}}$ in the $\mathcal{F}_{\text{PRG}}$-hybrid model.

**Proof.** Note that $P_Q$ receives no messages in the protocol, and hence the sharing protocol is trivially secure against corruption of $P_Q$. The only message that $P_A$ receives is the value $[x]_1 - b$. However, given the security of PRG, $b$ is a random value unknown to $P_A$. Thus, the distribution of $[x]_1 - b$ is uniformly random from $P_A$'s view and the information learned by $P_A$ can be perfectly simulated. Next, during the protocol, the server learns $[c]_1$ and $w + a$. However, the distribution of $[c]_1$ and $w + a$ is uniformly random from the server's view, since given the security of PRG, $a$ and $[c]_1$ are random values unknown to the server. Thus, the information learned by the server can be perfectly simulated. $\qquad\square$

**Theorem 3.** $\Pi_{\text{DReLU}}$ securely realizes the functionality $\mathcal{F}_{\text{DReLU}}$ in the $\mathcal{F}_{\text{PRG}}$-hybrid model.

**Proof.** Note that $P_A$ receives no messages in the protocol, and hence the sharing protocol is trivially secure against corruption of $P_A$. Next, the messages that the server receives are $[z]_0 - \delta$ and $\text{sign}(z) - \delta'$. However, given the security of PRG, $\delta$ and $\delta'$ are random values unknown to the server. Thus, the distribution of $[z]_0 - \delta$ and $\text{sign}(z) - \delta'$ is uniformly random from the server's view and the information learned by the server can be perfectly simulated. Then, the message that $P_Q$ learns is $z = rx$, where $r$ is a random number in the ring. Thus, the distribution of $z$ is uniformly random from the server's view. $\qquad\square$

**Theorem 4.** $\Pi_{\text{ReLU}}$ securely realizes the functionality $\mathcal{F}_{\text{ReLU}}$ in the $(\mathcal{F}_{\text{Matmul}}, \mathcal{F}_{\text{DReLU}})$-hybrid model.

**Proof.** Note that as shown in Figure 4, $\Pi_{\text{ReLU}}$ consists of $\Pi_{\text{DReLU}}$ and $\Pi_{\text{Matmul}}$. Therefore, the ReLU protocol is trivially secure in the $(\mathcal{F}_{\text{Matmul}}, \mathcal{F}_{\text{DReLU}})$-hybrid model. $\qquad\square$

**Theorem 5.** $\Pi_{\text{MaxPool}}$ securely realizes the functionality $\mathcal{F}_{\text{MaxPool}}$ in the $\mathcal{F}_{\text{ReLU}}$-hybrid model.

**Proof.** Note that as shown in Figure 4, $\Pi_{\text{MaxPool}}$ consists of $\Pi_{\text{ReLU}}$. Therefore, the MaxPool protocol is trivially secure in the $\mathcal{F}_{\text{ReLU}}$-hybrid model. $\qquad\square$

**Theorem 6.** $\Pi_{\text{Agg}}$ securely realizes the functionality $\mathcal{F}_{\text{Agg}}$ in the $\mathcal{F}_{\text{PRG}}$-hybrid model.

**Proof.** Note that $P_A$ receives no messages in the protocol, and hence the aggregation protocol is trivially secure against corruption of $P_A$. Next, the only message that the server receives is the value $[x_j]_0 - r_j$. However, given the security of PRG, $r_j$ is a random value unknown to the server. Thus, the distribution of $[x_j]_0 - r_j$ is uniformly random from the server's view and the information learned by the server can be perfectly simulated. After the aggregation, $P_Q$ only learns the aggregated result

Table 7: Experiment setting of different datasets during local model training.

|  | Loss Function | Learning Rate | Batch Size | Epoch | Number of Private Data |
|---|---|---|---|---|---|
| SVHN | cross-entropy | 0.5 | 256 | 250 | 1465 |
| CIFAR10 | cross-entropy | 0.1 | 64 | 500 | 1000 |
| Tiny ImageNet | cross-entropy | 0.01 | 64 | 500 | 10000 |

$\sum_{j\in[n]} x_j$, but is unknown to a single $x_j$. Therefore, the aggregation protocol is secure assuming the aggregation result will not reveal privacy.

□

## F  ADDITIONAL EXPERIMENTAL SETUP

**Datasets.** We evaluate PrivHFL on the following standard datasets for image classification:

- **SVHN.** SVHN is a real-world image dataset obtained from house numbers in Google Street View images. Each sample is 32×32 RGB image. We use 73257 digits for training and 26032 digits for testing.

- **CIFAR10.** CIFAR-10 consists of 60,000 32×32 RGB images in 10 classes. There are 50,000 training images and 10,000 test images.

- **Tiny ImageNet.** Tiny ImageNet contains 100,000 images of 200 classes (500 for each class) downsized to 64×64 colored images. Each class has 500 training images, 50 validation images and 50 test images.

**Model architecture.** We use VGG-7, ResNet-8 and ResNet-10 for SVHN and CIFAR10 datasets. VGG-7 contains 6 convolutional layers, followed by a fully-connected layer, where ReLU is applied as the activation function and MaxPooling is used for downsampling, giving 7 layers in total. ResNet-8 we use follows the same architecture in CaPC, which consists of a convolutional layer, 2 residual blocks with 4 convolutional layers, followed by a fully-connected layer. Compared with the original architecture, the last block is excluded and neurons of the last layer is increased. ResNet-10 begins with a convolutional layer, followed by 4 residual blocks with 2 convolutional layers in each block, and one final fully-connected layer, thus there are 10 functional layers overall. Besides, we use ResNet-14, ResNet-16 and ResNet-18 (He et al., 2016) for Tiny ImageNet.

**Training procedure.** Utilizing the private data, each client first trains the local model from scratch using SGD optimizer, where the detail training setting is shown in Table 7. Next, the clients run the PrivHFL protocol to generate query-response pairs, which will be used to retrain the local model. When retraining the models, they use Adam optimizer for 50 epochs with learning rate of 2e-3 decayed by a factor of 0.1 on 25 epochs, where the batch size is 256 on SVHN, and 64 on CIFAR10 and Tiny ImageNet.

## G  VISUALIZATION OF MIXUP AND SECRET-SHARING SAMPLES

The mixup-synthetic data expansion method of PrivHFL is only to expand private data locally to generate a large data pool, which cannot provide any privacy guarantee. Figure 19 shows the mixup images synthesized from the same pair of real images with different mixup coefficients $\lambda$. As shown in the figure, when $\lambda = 1$, the mixup image will be identical to the real sample. Even if $\lambda = 0.7, 0.5, 0.2$, the original images can still be visually recognized.

To further protect the $P_Q$'s privacy, the mixup images are *secret-shared* between the server and $P_A$, rather than being disclosed directly to them. As shown in Figure 20, the secret-shared images look like two random noises, since in secret sharing each share is randomly sampled from the ring $\mathbb{Z}_{2^{64}}$. To be specific, for a mixup image $x$, the server owns $x_1$, in which each element is randomly sampled from the ring $\mathbb{Z}_{2^{64}}$. Meanwhile, the answering party $P_A$ owns a random $x_0$ with $x_0 = x - x_1 \in \mathbb{Z}_{2^{64}}$. Besides, note that in our secure querying protocol, the intermediate values always maintain such *secret sharing invariant*, so that neither party can steal any private information about real datasets, local models and prediction results from the knowledge they obtain.

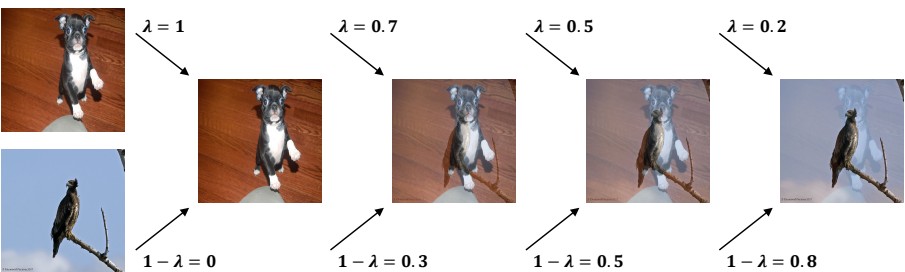

Figure 19: **Different mixup-synthetic images from the same pair of the natural images by varying the mixup coefficient** $\lambda$.

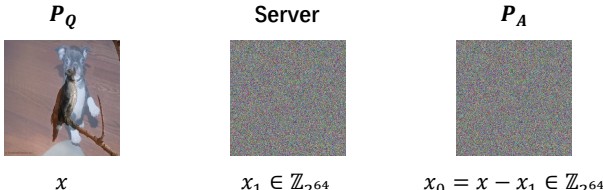

Figure 20: **An example of secret sharing on the mixup image.** $x$, $x_1$ and $x_0$ are pixel matrices, owned by the querying party $P_Q$, the server, and the answering party $P_A$, respectively.

