# OpenReview forum: "Practical and Private Heterogeneous Federated Learning"
_ICLR.cc/2022/Conference — ICLR 2022 Submitted_

### Official Review · Reviewer_kXk2 · 2021-11-01

**Correctness:** 2
**Technical Novelty And Significance:** 2
**Empirical Novelty And Significance:** 2
**Recommendation:** 5
**Confidence:** 4

**Main Review:**

**Strong points:**

1. The improvement in efficiency against the previous methods, e.g., CryptGPU or CaPC.

2. Good design. PrivHFL builds on top of CrytpGPU to compute over integer values (64 bit ring elements). CryptGPU embeds the integer-valued cryptographic operations into floating-point arithmetic.

**Weak points:**

1. **Private data revealed in plain text if $P_Q$ colludes with the server**. The revealed data is an augmentation of the private dataset using MixUp but it still leaks information about the private data. In contrast, e.g., in CaPC, once the server (the 3-rd trusted party - Privacy Guardian) and answering parties collude, only the predicted label is revealed but not the private data.

2. **The logits are aggregated** (as described in Figure 5) but these values are not normalized across models so each model can have a very different impact on the final predictions. This makes the method more vulnerable to attacks.

3. **MixUp was already used for Federated Learning in [1]**.  Yoon et al. propose to apply MixUp to Federated Learning for addressing the challenge of non-iid data.

4. MixUp is not a state-of-the-art (SOTA) data augmentation method. Could MixMatch [2] (it also shows improvements for MixUp) or FixMatch [3] or other SOTA be used? The other data augmentation methods from Figure 10 are rather rudimentary.

5. **No DP provided**. The revealed logits leak information about private data from the answering parties ($P_A$).

6. **Intermediate representations are released to $P_Q$**. From Section 3.3 "After that, $P_A$ and the server disclose the secret-shared value [z] to $P_Q$, which then computes the sign of z, i.e., the sign of x. sign(z) is shared to the server and $P_A$ by using the protocol in Figure 2." Figure 4 shows that scaled intermediate results (inputs to the ReLU layers): $z = r([x]_1 + [x]_2)$ are revealed to the querying party ($P_Q$). How much can $P_Q$ infer about the model from these intermediate results? It seems that $P_Q$ can infer the model architecture that is used by $P_A$. Is z some scaling of the original intermediate representation, or every value from the intermediate representation can be scaled differently to obtain the representation sent to the querying party. The latter might make the method practically robust since the querying party is rather not able to infer the parameters of the model used by the answering party. However, this should be confirmed formally. In the former case (i.e., z is only a scaled intermediate representation - input to ReLU), the parameters of the model used by $P_A$ might be extracted from z. However, it is claimed at the end of Section 3.3 that: "We give a formal security proof in Appendix E. Intuitively, PrivHFL reveals zero information to the answering parties $P_A$ and the server, and only reveals the final aggregated prediction to the querying party $P_Q$, since all intermediate values are secret-shared. Given the above, a corrupted $P_A$ cannot learn anything about the query data of querying parties, while the confidentiality of responding parties’ model parameters against corrupted $P_Q$ is also protected."

7. **No assumption on a public dataset**. For example, CaPC does not assume that there is a public dataset, thus the following is not correct: “Besides, to the best of our knowledge, **existing methods all require a public dataset** to implement heterogeneous federated learning.” Since the data is always encrypted in CaPC, the querying party can send the private data with a guarantee that it is not revealed to any other party. Additionally, CaPC can serve as a private consultation protocol. The natural new data to be labeled come from new patients or clients (in the case of hospitals or banks, respectively).

8. **No direct communication between clients.** This is easy to fix. For instance, CaPC can be extended to a setting where there is no direct communication between clients. If we have parties P1, P2, P3. P1 and P2, as well as P1 and P3 can communicate directly. They just need to encrypt their messages and ask P1 to pass them along.

9. CaPC is a modular protocol. CryptGPU, which has been published recently (after CaPC), could replace HE-transformer to accelerate the private inference while maintaining CaPC's strong privacy and confidentiality guarantees.

10. Active learning was also used in CaPC.

**Questions:**
1. Why is scalability an issue (in terms of the number of clients and size of the models)? Could you elaborate on it, please?
Figure 6: How many models have which architecture?
Table 1: For “Private data” is the assumption for the baseline that there are not labels for the private data and they are obtained by querying the answering parties?
2. How is the whole private inference implemented across all the layers? PrivHFL does not provide the source code. It would be informative to see how really the additive sharing is used for private inference and when the intermediate results are revealed to the querying party $P_Q$.
3. Should not DReLU be defined as: $\frac{1}{2}*(1 + sgn(x))$ ?

**Comments:**
1. Section D.1 is a repetition of the introduction.

**Typos:**

inferior than -> inferior to (Section 3.2)

experiment setup -> experimental setup (Section 4.1)

they "are" most uncertain (Section C.1.)

Suggestion: "number of private data" -> "number of private data samples"

they using -> they use (Section F)

\citep instead of \cite or \citet should be used in most cases (e.g., the first paragraph in the Introduction).

**References:**

[1] FedMix: Approximation of MixUp under Mean Augmented Federated Learning. Tehrim Yoon, Sumin Shin, Sung Ju Hwang, Eunho Yang. (ICLR 2021).

[2] MixMatch: A Holistic Approach to Semi-Supervised Learning. David Berthelot, Nicholas Carlini, Ian Goodfellow, Nicolas Papernot, Avital Oliver, Colin Raffel. arxiv:1905.02249 (NeurIPS 2019)

[3] FixMatch: Simplifying Semi-Supervised Learning with Consistency and Confidence. Kihyuk Sohn, David Berthelot, Chun-Liang Li, Zizhao Zhang, Nicholas Carlini, Ekin D. Cubuk, Alex Kurakin, Han Zhang, Colin Raffel.  (NeurIPS 2020)


**Summary Of The Paper:**

PrivHFL is a protocol for collaborative learning that does not reveal the private data to the server or answering parties $P_A$ if they operate as honest-but-curious entities. PrivHFL tackles the bottleneck of private inference by building a new protocol from scratch that can run on GPU. It is not assumed that parties have new data or that there is a public pool of data to run inference on but the additional query data for knowledge transfer is obtained by augmenting the private data with MixUp.

**Summary Of The Review:**

The paper shows much better performance (e.g., much faster execution of the private inference) but this is at the cost of lowering the security level. Additionally, the aggregation through logits makes the method more vulnerable to attacks than other proposed protocols, such as, CaPC. MixUp and Active Learning methods were already used in federated learning (FedMix, CaPC) so this is cannot be classified as a technical novelty.

---

> ### Author Response · Authors · 2021-11-19
> **Response to Reviewer kXk2 (1/3)**
>
> Dear Reviewer,
>
> We thank the reviewer for the thoughtful review and helpful comments.  We address each of your points below.
>
> ***Q1.*** *Private data revealed in plain text if $P_Q$ colludes with the server.*
>
> **A1. As shown in our threat model, PrivHFL assumes that the server and clients do not collude. The security of all protocols is built under this threat model. Therefore, this work (as well as almost all other works) does not consider any attackers who have more powerful capabilities outside of the defined threat model.** This security assumption is the same as that of CaPC (in Section 3.1). Similarly, CaPC did not discuss the impact of collusion since it is meaningless under the non-collusion threat model.
>
> ***Q2.*** *The logits that are not normalized can have a very different impact on the final predictions, and make the method more vulnerable to attacks.*
>
> **A2. In PrivHFL, it’s unnecessary to normalize the logits.** PrivHFL works in an honest-but-curious adversary setting, where each entity strictly follows the specification of designed protocols without breaching its integrity. Therefore, they can only attempt to infer private information through the obtained messages, and will not launch various attacks by modifying messages (e.g., logits). In addition, in the revision, we have added an ablation experiment in Figure 13 to further verify the impact of normalization on the model accuracy. The results show that, in PrivHFL, models can achieve better performance when using unnormalized values. What's more, the normalization operation can be easily implemented in the secure computation, but this will introduce extra costs. Therefore, in PrivHFL, the clients do not normalize logits. Please refer to Appendix B.3 for more details.
>
> ***Q3.*** *MixUp was already used for Federated Learning in [1].*
>
> **A3.** Thanks for bringing [1] to our attention. In the revision, we outline the difference between our work and [1] in footnote 7 in Appendix D.1. Specifically, in our work, the role of mixup is to expand the private data to generate a large unlabeled query data pool, whereas in [1] mixup is used to address the issue of non-iid data. Note that our dataset expansion can be seen as a general framework to solve the problem of existing works relying on auxiliary datasets, and mixup is a concrete technique to instantiate this framework. Other recent data augmentation methods, such as cutmix [2] and cutout [3], may also be good choices for the data expansion, and in the revision, we have added experiments on these methods in Figure 11(c).
>
> ***Q4.*** *Could MixMatch or FixMatch or other SOTA be used?*
>
> **A4. MixMatch and FixMatch cannot be directly applied to PrivHFL, whereas other SOTA data augmentation methods may be effective.** MixMatch and FixMatch provide effective methods of leveraging unlabeled data to improve models' performance in the semi-supervised learning. However, their methods cannot be directly applied to PrivHFL for data expansion, because unlabeled data samples do not exist in our work. Notably, the data expansion of PrivHFL is a general framework to solve the problem of auxiliary datasets, and the mixup technique provides one (but not the only) instantiation. In fact, due to its modular design, PrivHFL can be readily extended with SOTA data augmentation techniques. In the revision, we have also added experiments in Figure 11(c) to verify the effects of recent techniques, including cutmix [2] and cutout [3] (Note that cutout is also used in FixMatch).
>
> ***Q5.*** *No DP provided.*
>
> **A5. The DP guarantee can complement our work.** In the revision, we have added more descriptions about the DP extension in Section 3.4. A well-designed DP mechanism can be used as a plug-and-play module to prevent privacy leakage from the aggregated results. However, this is non-trivial to design a customized DP mechanism for HFL, because the privacy-utility tradeoff is difficult to resolve. Especially, the privacy guarantee will deteriorate with the increase of corrupted clients, unless it is mitigated by adding more DP noise, but at the cost of comprising accuracy. In our setting, we assume up to $n-1$ ($n$ is the number of clients) clients can be corrupted such that the above problem will be escalated to the worst case. The same issue is also encountered by CaPC. Therefore, it is an interesting and challenging work to design a high-utility DP mechanism in the distributed scenario where multiple clients may be corrupted.
>
> [1] Tehrim Yoon, Sumin Shin, Sung Ju Hwang, Eunho Yang. “FedMix: Approximation of MixUp under Mean Augmented Federated Learning”. ICLR, 2021.
>
> [2] Sangdoo Yun, Dongyoon Han, Seong Joon Oh, Sanghyuk Chun, Junsuk Choe, and Youngjoon Yoo. "Cutmix: Regularization strategy to train strong classifiers with localizable features." ICCV, 2019.
>
> [3] Terrance, and Graham W. Taylor. "Improved regularization of convolutional neural networks with cutout." arXiv:1708.04552, 2017.

---

> > ### Author Response · Authors · 2021-11-19
> > **Response to Reviewer kXk2 (2/3)**
> >
> > ***Q6.*** *Intermediate representations are released to $P_Q$.*
> >
> > **A6. 1) In our protocol, each $z$ is the scaling of different values of $r$, which is necessary for security.** In the revision, we have modified the description of the ReLU protocol in Section 3.3 to make it clearer. Note that we do not disclose any information (except the size) about the model parameters, and only reveal the masked values (rather than the plain values) of intermediate features, i.e., $z = rx$, where $x$ is the feature maps of the operated layer and $r$ is a number randomly sampled from the ring $\mathbb{Z}_{2^{64}}$. Similar ideas are also applied to other privacy-preserving machine learning works [4] [5]. Besides, we also formally prove its security in Theorem 3 of Appendix E. **2) Almost all privacy-preserving machine learning based on MPC cannot prevent the leakage of model architectures.** Similarly, in PrivHFL, the model architecture of $P_A$ is also leaked to the server and $P_Q$ since the secret sharing technique can only hide the value of the original data, not the size. Therefore, the leakage is not just because the masked intermediate representations are exposed to $P_Q$.
> >
> > ***Q7.*** *No assumption on a public dataset in CaPC.*
> >
> > **A7. Thanks for pointing this out. We have revised the statement from “public datasets” to “auxiliary datasets” in the revision.** The main goal of PrivHFL is to achieve a practical setting in heterogeneous federated learning that abandons the dependence on any auxiliary datasets, including public datasets in prior works and private unlabeled datasets employed in CaPC. This setting may be more desirable, especially in the scenarios where data are rare and data collection is difficult, like rare disease diagnosis.
> >
> > ***Q8.*** *CaPC can be extended to the setting where there is no direct communication between clients.*
> >
> > **A8. By carefully designing protocols, CaPC can be extended to the communication-limited setting but at the cost of communication overhead.** Note that the original CaPC protocol is not compatible with communication restrictions between clients. In the revision, we carefully and non-trivially design two modified CaPC protocols for this setting in Appendix C.5. However, as analyzed, such extension comes at the cost of increased communication overhead. Therefore, given that PrivHFL has outperformed the original CaPC, PrivHFL has better advantages compared to the modified protocols.
> >
> > ***Q9.*** *CaPC is a modular protocol and CryptGPU could replace HE-transformer in CaPC.*
> >
> > **A9. CryptGPU cannot be directly extended to CaPC in the scenario of limited communication that we consider.** This is because CryptGPU is a three-party protocol that requires communication between any pair of clients. In PrivHFL, we just exploit GPU-accelerated multiplication under the secret-sharing form provided by CryptGPU, but all cryptographic protocols are redesigned from scratch. These protocols are suitable for our scenario, where client communication is restricted and batch prediction is required. Besides, PrivHFL is also friendly to secure predictions under CPUs, where GPU-accelerated multiplication is not employed. As shown in Table 3, even under the CPU setting, the performance of PrivHFL is still better than CryptGPU.
> >
> > ***Q10.*** *Active learning was also used in CaPC.*
> >
> > **A10.** Indeed. We follow CaPC and use active learning strategies to encourage the querying party to actively select query data. In our revision, we have revised the description in Section 3.2 and Appendix C.3. Note that active learning is not the main technical and empirical novelty in PrivHFL. All experiments in the main text are based on random sampling and do not rely on any active learning strategies. Besides, as shown in Figure 12, active learning is not the main factor to improve the accuracy of heterogeneous models in PrivHFL.
> >
> > [4] Sameer Wagh, Divya Gupta, and Nishanth Chandran. "SecureNN: 3-Party Secure Computation for Neural Network Training." PET, 2019.
> >
> > [5] Liyan Shen, Xiaojun Chen, Jinqiao Shi, Ye Dong, and Binxing Fang. "An Efficient 3-Party Framework for Privacy-Preserving Neural Network Inference." ESORICS, 2020.

---

> > > ### Author Response · Authors · 2021-11-19
> > > **Response to Reviewer kXk2 (3/3)**
> > >
> > > ***Q11.*** *Why is scalability an issue?*
> > >
> > > **A11. In the revision, we have added more descriptions about the scalability in Section 3.4.** Specifically, in PrivHFL as well as general heterogeneous federated learning, each client can play the role of the querying party and the answering party at the same time in the querying phase. This will incur the overhead of $O(n^2)$ secure predictions for each iteration, where $n$ is the number of clients. As the size of the model or the number of clients increases, such overhead issue (especially communication cost) will be the main bottleneck for the scalability of the system.
> > >
> > > ***Q12.*** *Figure 6: How many models have which architecture?*
> > >
> > > **A12. We have added the experimental setup in Section 4.1 of our revised paper.** In Figure 6, we follow the CaPC's heterogeneous setting and use VGG-7, ResNet-8 and ResNet-10 architectures for each $n/3$ clients, where $n=50$ is the number of clients.
> > >
> > > ***Q13.*** *Table 1: For “Private data”, is the assumption for the baseline that there are not labels for the private data and they are obtained by querying the answering parties?*
> > >
> > > **A13. In our baseline, private data refer to the client's training dataset including labels**, which is the only knowledge that the client obtains, since we do not assume any auxiliary datasets. In the revision, we have added more descriptions about the details of the private data baseline in Section 3.2. The reason for querying with labeled training data is that the queried soft labels have more information to assist training. The experimental results also confirmed this intuition. Specifically, we can view the querying procedure as knowledge distillation. After querying, $P_Q$ (i.e. the student) obtains the aggregated logit and hence the soft label from the ensemble of $P_A$'s (i.e., the teacher). Knowledge distillation shows that the soft labels are well-informed and help guide the optimization of the $P_Q$'s model.
> > >
> > > ***Q14.*** *How is the whole private inference implemented across all the layers?*
> > >
> > > **A14. In the revision, we have added a graphic depiction in Figure 17 to explain the cross-layer implementation of the whole private inference.** The inputs and outputs of each layer are secret-shared between $P_A$ and the server. Thus, by maintaining such secret sharing invariants, the secure prediction can be implemented layer by layer. In Appendix C.2, we give a more detailed analysis. Besides, in the whole inference, all intermediate values are secret-shared, except for opening a masked value $z$ (rather than the plain value as described in A6) to $P_Q$ in the DReLU operation. In addition, it's worth emphasizing that the better performance of PrivHFL is not caused by lowering the security level, but by the fact that secret sharing abandons expensive cryptographic evaluation such as HE and GC. More importantly, we design efficient secure protocols based on secret sharing for linear and non-linear layers, which greatly improves the performance of private inference.
> > >
> > > ***Q15.*** *Should not DReLU be defined as: $1/2 \star (1+sgn(x))$?*
> > >
> > > **A15. The sign(x) function defined in PrivHFL is the most significant bit (MSB) of the value x.** In the revision, we have added its definition in Section 3.3. Specifically, in the secure computation filed, each value is represented in two’s complement, and hence the sign bit is the most significant bit (MSB).
> > >
> > > ***Q16.*** *Section D.1 is a repetition of the introduction.*
> > >
> > > **A16. Thanks for pointing this out. We have modified Section D.1 carefully.**
> > >
> > > ***Q17.*** *Typos.*
> > >
> > > **A17. We have corrected the typos and double-checked our paper carefully.**

---

> > > > ### Comment · Reviewer_kXk2 · 2021-11-27
> > > > **Scalability**
> > > >
> > > > What is the running time of PrivHFL when the number of parties $n$ is set to 5, 10, 25, 50, 100, 250?

---

> > > > > ### Author Response · Authors · 2021-11-27
> > > > > **Response to Reviewer kXk2 (2/2)**
> > > > >
> > > > > ***Q8.*** *Scalability*
> > > > >
> > > > > **A8.** In our current experimental evaluation, we only have results when the number of clients is 10, 30, 50. But it can be estimated for 100 and 250, because the runtime of *each client* in the query process is linear with the number of clients. Therefore, the table below shows the actual runtime (second) and the estimated runtime (* denotes estimated values) on CIFAR10 under 2500 query data. As shown in the table, each query takes about 21 minutes on ResNet10 over 250 clients. In summary, the scalability of the system is limited by the number of clients.
> > > > >
> > > > > |#clients|VGG7|ResNet8|ResNet10|
> > > > > |:--------------------:|:--------------------:|:--------------------:|:--------------------:|
> > > > > |10|28.24|43.95|55.41|
> > > > > |30|81.12|127.52|158.33|
> > > > > |50|133.52|206.20|256.39|
> > > > > |100*|267.04*|412.40*|512.78*|
> > > > > |250*|667.60*|1031.00*|1281.95*|

---

> > > > > > ### Comment · Reviewer_kXk2 · 2021-11-29
> > > > > > **Scalability**
> > > > > >
> > > > > > Thank you for your responses. In the previous answer the authors wrote:
> > > > > >
> > > > > > >**This will incur the overhead of $O(n^2)$ secure predictions for each iteration, where $n$ is the number of clients. As the size of the model or the number of clients increases, such overhead issue (especially communication cost) will be the main bottleneck for the scalability of the system.**
> > > > > >
> > > > > > However, now they claim:
> > > > > >
> > > > > > >**the runtime of each client in the query process is linear with the number of clients**
> > > > > >
> > > > > > Why is there this discrepancy?
> > > > > >
> > > > > > CaPC (Section 4.2) follows the papers on PATE [1,2] and uses 250 models for SVHN. However, in this paper it is written:
> > > > > >
> > > > > > >**For SVHN and CIFAR10, following CaPC we set the number of clients n = 50**
> > > > > >
> > > > > > Another interesting aspect would be to show the performance of PrivHFL as the model size increases, e.g., for ResNet8, ResNet10, ..., ResNet18, ResNet34, and so on.
> > > > > >
> > > > > > For such work as PrivHFL, it is recommended to submit the source code.
> > > > > >
> > > > > > **References:**
> > > > > >
> > > > > > [1] [Semi-supervised Knowledge Transfer for Deep Learning from Private Training Data.](https://openreview.net/forum?id=HkwoSDPgg) Nicolas Papernot, Martín Abadi, Úlfar Erlingsson, Ian Goodfellow, Kunal Talwar. ICLR 2017.
> > > > > >
> > > > > > [2] [Scalable Private Learning with PATE.](https://openreview.net/forum?id=rkZB1XbRZ) Nicolas Papernot, Shuang Song, Ilya Mironov, Ananth Raghunathan, Kunal Talwar, Úlfar Erlingsson. ICLR 2018.

---

> > > > > > > ### Author Response · Authors · 2021-11-29
> > > > > > > **Response to Reviewer kXk2**
> > > > > > >
> > > > > > > Dear Reviewer,
> > > > > > >
> > > > > > > Thanks for your valuable feedback.
> > > > > > >
> > > > > > > ***Q1.*** *Scalability*
> > > > > > >
> > > > > > > **A1.** Thanks for pointing this out. For all clients, the PrivHFL system will incur $O(n^2)$ secure predictions, whereas in the table of the above answer, we report the cost of each client (i.e., $O(n)$ secure predictions) during the query process.
> > > > > > >
> > > > > > > ***Q2.*** *SVHN*
> > > > > > >
> > > > > > > **A2.** **Thanks for pointing this error out. We will modify this description to make it correct.** In our evaluations, we set the number of clients $n = 50$ for both SVHN and CIFAR10.  For SVHN, we use 73,257 training data, but CaPC additionally uses the extra training data and hence sets $n=250$. As stated in their paper, "the number is larger for SVHN because we have more data".
> > > > > > >
> > > > > > > ***Q3.*** *Performance as the model size increases*
> > > > > > >
> > > > > > > **A3.** **We will add more experimental evaluations as the model size increases. In the current revision, we have partially shown the influence of different model sizes on the runtime.** For instance, as shown in Table 2, we have reported the runtime of secure querying over CIFAR10 and SVHN for VGG7, ResNet8 and ResNet10 with 50 clients, while we have reported the runtime over Tiny ImageNet for ResNet14, ResNet16 and ResNet18 with 10 clients. Moreover, we also reported the runtime of secure prediction over CIFAR10 for VGG7, VGG9, VGG11, ResNet8, ResNet10 and ResNet12 in Figure 8.
> > > > > > >
> > > > > > > ***Q4.*** *Source code*
> > > > > > >
> > > > > > > **A4.** The code will be provided with the final version and will be open-sourced.

---

> > ### Comment · Reviewer_kXk2 · 2021-11-27
> > **MixUp, Privacy, Collusion, Diffie-Hellman, and Normalization**
> >
> > Thank you for the answers.
> >
> > >**MixUp is a very well-known method and has already been used for Federated Learning in prior work [1].**
> >
> > In the main contributions it is overclaimed: *We design a simple yet effective dataset expansion method [...])*. This is confusing since MixUp is not designed in this paper but simply used. If the usage of MixUP as in Yoon et al. is added only to the footnote, it is rather not enough as we see [here](https://openreview.net/forum?id=pIjvdJ_QUYv&noteId=hfj7WJwHjSl).
> >
> > >**CutMix**
> >
> > Thank you for comparing the methods. The differences between different augmentations are rather small here.
> > On the other hand, CutMix consistently achieves significant enhancements across different tasks. [2] These 3 methods are significantly better than previously considered ones (Gaussian Random Noise or Random Flipping).
> >
> > >**However, this is non-trivial to design a customized DP mechanism for HFL, because the privacy-utility tradeoff is difficult to resolve.**
> >
> > How would you design a new DP mechanism? If only labels are revealed, then PATE can be used as in CaPC and no new DP mechanism has to be designed from scratch.
> >
> > >**Lower privacy than in CaPC.**
> >
> > CaPC provides privacy since only the final label is revealed to the querying party via PATE.
> >
> > In PrivHFL there is no privacy of the training data and the aggregated logits, which contain much more information than a label, are returned to the querying party.
> >
> > >**Collusion**
> >
> > If collusion is considered, then CaPC is more secure since only labels are leaked and not the entire data samples.
> >
> > >**Diffie-Hellman**
> >
> > *$Sk_{QA}$ is constructed by using the Diffie-Hellman key agreement protocol and setting up the server as an intermediary for communication.* The Diffie-Hellman key exchange is vulnerable to a man-in-the-middle attack as explained [here](http://security.nknu.edu.tw/crypto/faq/html/3-6-1.html).
> >
> > >**Normalization**
> >
> > What kind of normalization is used in Figure 13? Are the softmax values used there?

---

> > > ### Author Response · Authors · 2021-11-27
> > > **Response to Reviewer kXk2 (1/2)**
> > >
> > > ***Q1.*** *MixUp is a very well-known method and has already been used for Federated Learning in prior work [1].*
> > >
> > > **A1.**  1) **Although mixup is already a well-known regularization technique, we are the first to extend it to Heterogeneous Federated Learning as a dataset expansion method.** In the Introduction, we have clearly stated that we simply used the mixup technique. 2) In the subsequent revision, we will add a section specifically to discuss the difference from the prior work [1].
> > >
> > > ***Q2.*** *CutMix*
> > >
> > > **A2.** Indeed, these three data augmentation methods show similar performance and are significantly better than previously considered ones. **Note that our dataset expansion can be seen as a general framework, and mixup, cutout and cutmix can all be used as the concrete techniques to instantiate this framework.** Please refer to Section 3.2 for more details.
> > >
> > > ***Q3.*** *It's non-trivial to design a customized DP mechanism for HFL, because the privacy-utility tradeoff is difficult to resolve.*
> > >
> > > **A3. A new DP mechanism is needed to solve the problem that multiple clients may be corrupted.** As explained by CaPC, "this setup (i.e., the PATE-based DP mechanism) assumes that only one answering party can be corrupted. If instead $C$ parties are corrupted, the sensitivity of the noisy aggregation mechanism will be scaled by $C$ and the privacy guarantee will deteriorate". The main way to alleviate this privacy issue is to add more DP noises, but at the cost of comprising accuracy. Especially when $n-1$ (out of $n$) clients are corrupted, this problem will escalate to the worst case. Therefore, we call for a new DP mechanism to be designed to solve this problem.
> > >
> > > ***Q4.*** *Lower privacy than in CaPC.*
> > >
> > > **A4. Compared to CaPC, our scheme may expose more information through the aggregated logits instead of the training data.** Under our threat model, the training data is completely hidden in the form of secret sharing. Note that in contrast to CaPC, we do not rely on any unlabeled problem-domain datasets but use synthetic datasets, and hence we cannot directly use the queried labels to improve the model. As a result, the use of the aggregated logits actually requires a compromise with rigorous privacy guarantees.
> > >
> > > ***Q5.*** *Collusion*
> > >
> > > **A5. Indeed, PrivHFL leaks the data samples if collusion is considered.** However, as stated in the above rebuttal, the security of all cryptographic protocols is built under their specific threat model (PrivHFL/CaPC assumes that the server and clients do not collude). The non-colluding threat model is a very common setting in the privacy-preserving federated learning [2], the secure outsourcing data preprocessing [3] and model training [4]. If collusion is considered, [2] will leak the model gradients, [3] will leak the entire data samples and [4] will also leak the entire model parameters and the intermediate values.
> > >
> > > ***Q6.*** *Diffie-Hellman*
> > >
> > > **A6.** As also explained [here](http://security.nknu.edu.tw/crypto/faq/html/3-6-1.html), the authenticated Diffie-Hellman key agreement protocol or Station-to-Station protocol are solutions to defeat the man-in-the-middle attack, which can be directly extended to PrivHFL. Besides, PrivHFL works in an honest-but-curious adversary setting, where each entity strictly follows the specification of designed protocols, and hence the server will not maliciously launch such attack. Moreover, the Diffie-Hellman key agreement protocol is also used in existing privacy-preserving federated learning works [5] [6].
> > >
> > > ***Q7.*** *Normalization*
> > >
> > > **A7. We use $l_2$ normalization to process the logit predicted by each client. The softmax function is used on the aggregated logits during the retraining phase.**
> > >
> > > [1] Tehrim Yoon, Sumin Shin, Sung Ju Hwang, Eunho Yang. "FedMix: Approximation of MixUp under Mean Augmented Federated Learning." ICLR, 2021.
> > >
> > > [2] Chengliang Zhang, Suyi Li, Junzhe Xia, and Wei Wang."BatchCrypt: Efficient Homomorphic Encryption for Cross-Silo Federated Learning." USENIX ATC, 2020.
> > >
> > > [3] Xiling Li, Rafael Dowsley and Martine De Cock. "Privacy-Preserving Feature Selection with Secure Multiparty Computation." ICML, 2021.
> > >
> > > [4] Sameer Wagh, Divya Gupta, and Nishanth Chandran. "SecureNN: 3-Party Secure Computation for Neural Network Training." PET, 2019.
> > >
> > > [5] Keith Bonawitz, et al. "Practical Secure Aggregation for Privacy-Preserving Machine Learning." CCS, 2017.
> > >
> > > [6] Keith Bonawitz, et al. "Practical Secure Aggregation for Privacy-Preserving Machine Learning." CCS, 2017.

---

> ### Author Response · Authors · 2021-12-01
> **Response to Reviewer kXk2**
>
> Dear Reviewer,
>
> Thank you again for your valuable comments! Since the discussion period is approaching its end, we wonder if you have any further question. If so, please raise them and we're ready to clarify further any time.
>
> With respects,
>
> Authors

---

### Official Review · Reviewer_3JAx · 2021-11-02

**Correctness:** 3
**Technical Novelty And Significance:** 2
**Empirical Novelty And Significance:** 3
**Recommendation:** 6
**Confidence:** 3

**Main Review:**

Query data generation: It is not clear how $\lambda$ is selected?  Is there any theoretical justification for the claim that "this methods provides a good coverage of the manifold of natural samples"? Given two private samples, the authors generate multiple query samples. It is not clear how the private samples, $\lambda$, and the number of synthetic samples for each given pair of private samples should be selected to achieve the optimal coverage of natural distributions.

Query-data sharing:  It is not clear how $Sk_{QA}$ can be constructed without direct communication channels between clients and considering that each client can play the role of both querying party and responding party at the same time (possibly responding to multiple queries)?

Result aggregation: It is assumed that all clients participate. This is not the typical scenario in federated learning. In addition, a unique $r_j$ is generated for each client, which is not consistent with other steps of the algorithm.

Experiments: How much will the accuracy degrade if you set secret-sharing protocols over other rings?

Algorithm 1: Why do you need line 5 while each client already trained local models on its private dataset?

Minor comment:

On the RHS of Figure 4, $\delta'_i$ should be $\delta'$



**Summary Of The Paper:**

This paper introduces a heterogeneous federated learning framework without requiring public datasets by designing a dataset expansion method and constructing cryptographic protocols for secure prediction. The main strength of this paper is regarding the experimental results. The authors have provided experiments on different datasets, heterogeneous local models, various degree of non-IID-ness across clients, and showed the runtime of three steps in their proposed secure querying protocol.

**Summary Of The Review:**

This paper is well-written and provides extensive experiments which show the efficiency of PrivHFL and accuracy gains compared to a baseline, which uses models trained on the local datasets. Some aspects of the dataset expansion method and  the secure querying protocol are unclear, which require further clarifications.

---

> ### Author Response · Authors · 2021-11-19
> **Response to Reviewer 3JAx (1/2)**
>
> Dear Reviewer,
>
> Thanks for your valuable feedback. The detailed responses are summarized as follows.
>
> ***Q1.*** *It is not clear how $\lambda$ is selected?*
>
> **A1. We empirically set $\lambda$ as $\lambda \in [0.1, 0.9]$ with an interval of $0.1$ that facilitates to generate more diverse images.**
>
> ***Q2.*** *Is there any theoretical justification for the claim that "this method provides a good coverage of the manifold of natural samples"?*
>
> **A2. It is difficult to theoretically analyze the distribution of synthetic samples, but we justify this statement experimentally in Figure 11.** Specifically, as shown in Figure 11(a) and 11(b), we visualize the synthetic samples in the feature space using the t-SNE dimensionality reduction technique. Compared with the original samples, the synthetic samples cover a larger part of the feature space and hence they should be diverse and informative. Besides, in Table 1 and Figure 7, we observe that using more query samples in the synthetic pool is beneficial to improve the model accuracy. This also implicitly indicates the diversity of the synthetic samples.
>
> ***Q3.*** *It is not clear how the private samples, $\lambda$, and the number of synthetic samples for each given pair of private samples should be selected to achieve the optimal coverage of natural distributions.*
>
> **A3. We randomly sample private data from the private dataset to generate synthetic samples, where $\lambda$ is set to $0.1, 0.2, \cdots, 0.9$ and each pair of private data is used to generate nine synthetic samples.** To verify the influence of different $\lambda$ values, in our revision, we have added an ablation experiment in Table 5. The experimental results show that the accuracy difference of the query model under different $\lambda$ values is not significant. This is because mixup in this work is used to generate diverse samples, and the $\lambda$ value is not the main factor that affects the performance. The article mainly aims to provide a general dataset expansion method to relax the assumption of auxiliary datasets, rather than optimizing the hyperparameters to improve performance. Designing specific schemes and experiments to find the possibly optimal $\lambda$ is an interesting research direction as future work.
>
> ***Q4.*** *It is not clear how $Sk_{QA}$ can be constructed without direct communication channels between clients?*
>
> **A4. $Sk_{QA}$ is constructed by using the Diffie-Hellman key agreement protocol and setting up the server as an intermediary for communication.** In our revision, we clarify this issue in Figure 18 and Appendix C.4. Given that each client can play the role of both the querying party and the responding party at the same time, we generate seeds for each pair of clients. For example, with $n$ clients, each client (whether as the querying party and the responding party) holds $n-1$ seeds, each of which is constructed together with one of the other parties.
>
> ***Q5.*** *It is assumed that all clients participate. This is not the typical scenario in federated learning.*
>
> **A5. Not really. We only select a subset of responding parties to participate for each secure querying procedure, which is suitable to the typical scenario in federated learning.** The details can be found in our revised Algorithm 1. Besides, in Table 1, we conducted ablation experiments to explore the impact of different fractions of participating clients on PrivHFL.
>
> ***Q6.*** *Result aggregation: A unique $r_j$ is generated for each client, which is not consistent with other steps of the algorithm.*
>
> **A6. The unique $r_j$ for each client is to indicate that different masks are used to hide different clients’ private information, which is necessary for security.** In the revision, we have revised Figure 5 and the description in Section 3.3 to make it clearer.
>
> ***Q7.*** *How much will the accuracy degrade if you set secret-sharing protocols over other rings?*
>
> **A7. In the secure prediction, the model accuracy depends on the precision of fixed-point encoding, which in turn determines the size of the underlying ring.** As recommended by CryptGPU [1], we set the precision as 20 bits, which ensures negligible relative error for private inference on Tiny-ImageNet. (The relative error is defined as $|ACC_{cipher} - ACC_{plain}| / ACC_{plain}$.) As such, to prevent overflow in the arithmetic evaluation over secret-shared data, shares are represented by a 64-bit ring while a 32-bit ring is no longer sufficient. Moreover, another advantage of the 64-bit ring is that the modulo operation can be implemented using regular arithmetics on the long integer type with no extra cost.
>
> [1] Sijun Tan, Brian Knott, Yuan Tian, and David J. Wu. "CRYPTGPU: Fast Privacy-Preserving Machine Learning on the GPU." IEEE S&P 2021.

---

> > ### Author Response · Authors · 2021-11-19
> > **Response to Reviewer 3JAx (2/2)**
> >
> > ***Q8.*** *Algorithm 1: Why do you need line 5 while each client already trained local models on its private dataset?*
> >
> > **A8. Thanks for pointing this error out. We have modified this algorithm to make it correct in the revision.**
> >
> > ***Q9.*** *The contributions are only marginally significant or novel in the aspect of technical novelty and significance.*
> >
> > **A9. The most important contribution of this work lies in the proposed cryptographic protocols.** In general, in scenarios where clients can not directly communicate, it is challenging to design an efficient and GPU-compatible secure prediction scheme while supporting batch prediction. Although a lot of prior works have proposed secure predictions, they either cannot be directly extended to PrivHFL due to the communication issues, or achieve poor performance in batch prediction due to the use of heavy cryptographic operations (such as OT and HE). To solve this challenge, we design several secure protocols from scratch using the lightweight secret sharing primitive. The main design principles include: 1) using PRGs to solve the communication limitation between clients; and 2) avoiding the use of heavy cryptographic protocols, and designing vectorized GPU-compatible protocols. As a result, the resulting protocols outperform prior art by up to two orders of magnitude. In the revision, we highlight the challenges and contributions in Section 1.
> >
> > ***Q10.*** *Typos.*
> >
> > **A10. We have corrected the typos and double-checked our paper carefully.**

---

> > > ### Author Response · Authors · 2021-11-29
> > > **Response to Reviewer 3JAx**
> > >
> > > Dear Reviewer,
> > >
> > > Thanks again for your valuable comments! As the discussion period will end soon, could you please kindly check our responses and revisions? We believe that our responses and revisions have addressed some of your concerns.

---

> > > > ### Comment · Reviewer_3JAx · 2021-12-01
> > > > **Response**
> > > >
> > > > Thank you for your detailed response and clarifications. Most of my comments are addressed except the one regarding theoretical justification for the claim that "this method provides a good coverage of the manifold of natural samples".

---

> > > > > ### Author Response · Authors · 2021-12-02
> > > > > **Response to Reviewer 3JAx**
> > > > >
> > > > > Dear Reviewer,
> > > > >
> > > > > Thanks for the feedback.
> > > > >
> > > > > Since this paper mainly focuses on the privacy protection with cryptography, for the generalization of the mixup-based data expansion method, we only experimentally verified this argument without theoretical analysis. We agree that theoretical investigation is an interesting and important research direction. A recent work [1] theoretically proved that the mixup augmentation corresponds to a specific type of data-adaptive regularization to reduce overfitting. One possible idea is to follow the thoughts of [1] to prove that mixup can also improve the manifold coverage of natural samples. We will consider and explore this as future work.
> > > > >
> > > > > [1] Linjun Zhang, Zhun Deng, Kenji Kawaguchi, Amirata Ghorbani, and James Zou. "How Does Mixup Help with Robustness and Generalization?" ICLR, 2021.
> > > > >
> > > > > Best regards,
> > > > >
> > > > > Authors

---

### Official Review · Reviewer_ZGSK · 2021-11-02

**Correctness:** 2
**Technical Novelty And Significance:** 3
**Empirical Novelty And Significance:** 2
**Recommendation:** 6
**Confidence:** 2

**Main Review:**

The authors proposed to use synthetic dataset generated using the Mixup method. I may have the following questions regarding this approach:
1. How can we guarantee that no privacy information will be breached using this synthetic method?
2. Will there be possibility to reconstruct the original dataset from this simple mixture? For example, a mixup will be identical to a real sample when 𝛌 = 1. Will this still leak the privacy of local data?


**Summary Of The Paper:**

The paper showed a new approach for heterogeneous federated learning, which uses augmented dataset instead of a public dataset for knowledge transfer between heterogeneous models. The authors also suggested a lightweight additive secret sharing technique to construct a series of tailored cryptographic protocols, which is friendly with GPU and the CUDA kernels.
The method was evaluated in a simulated scenario with three public imaging datasets. And shows superiority over the baseline methods, and also efficiency with regard to run time and communication cost.


**Summary Of The Review:**

I feel not fully confident to assess the quality of the work as I am non-expert in federated learning.

---

> ### Author Response · Authors · 2021-11-19
> **Response to Reviewer ZGSK**
>
> Dear Reviewer,
>
> Thanks for your valuable feedback. We address your questions below.
>
> ***Q1.*** *How can we guarantee that no privacy information will be breached using Mixup synthetic method?*
>
> **A1.** **In PrivHFL, we use designed secure querying protocols rather than mixup to guarantee no privacy breach.** Specifically, mixup's role is only to expand the private data locally to generate a large unlabeled data pool. Then in the querying phase, the query data selected from this pool will be additively secret-shared between the server and the answering party. The secret-sharing invariant is maintained for all context (including model inputs, intermediate values, and outputs) during the entire querying phase. Therefore, neither party can infer any private information about origin datasets, local models and prediction results from the messages they obtain.
>
> ***Q2.*** *Will there be possibility to reconstruct the original dataset from this simple mixture?*
>
> **A2.** **Indeed. But as described in A1, mixup is not used for privacy. The privacy of local data is still guaranteed by the additive secret sharing primitive.** Specifically, in the revision, we have added Figure 19 to show samples with mixup. As shown in the figure, when $\lambda = 1$, the mixup image will be identical to the real sample. Even if $\lambda = 0.7, 0.5, 0.2$, the original image can still be visually recognized. Therefore, in PrivHFL, mixup does not play a role in protecting privacy. We utilize the additive secret sharing technique to protect the privacy of original samples. In our revision, we have added an example in Figure 20 to visualize the secret-shared results and have given the detailed analysis. As shown in the figure, the secret-shared pictures look like two random noises, since in the secret sharing, each share is randomly sampled from the ring $\mathbb{Z}_{2^{64}}$. Besides, Appendix E also gives the formal security analysis.
>
> ***Q3.*** *The contributions are only marginally significant or novel in the aspect of empirical novelty and significance.*
>
> **A3.** **We conducted abundant experiments in the aspect of model accuracy and protocol efficiency. To better verify the effectiveness of PrivHFL, in the version, we have added additional comparison and ablation experiments.**
> (1)	On the evaluation of model accuracy, we give an end-to-end illustration on the performance of PrivHFL in Figure 6, and conduct ablation experiments to explore the impact of different factors on PrivHFL, including the fraction of participating clients, the number of query data (in Table I and Figure 7), query data selection strategies (in Figure 12), the degree of Non-IID-ness (in Figure 15), and the number of private data (in Figure 16). On the evaluation of protocol efficiency, we give the runtime and communication cost of PrivHFL and compare it with SOTA works in Tables 3 and 4, including CyptFlow2 (CCS 2020), CryptGPU (IEEE S&P 2021), CaPC (ICLR 2021). In addition, we report the overhead of each step in PrivHFL in Table 2 and the runtime under CPU and GPU settings in Figure 8.
> (2)	In the revision, we have added an ablation experiment in Table 5, to verify the influence of different $\lambda$ values in the mixup-synthetic method. And we have conducted experiments on other data augmentation methods in Figure 11(c). Further, we have added Figure 13, to verify the impact of the normalization operation on the model accuracy.
> If there are any deficiencies in the experimental evaluation, please kindly point it out, and then we will make further improvements.

---

> > ### Author Response · Authors · 2021-11-29
> > **Response to Reviewer ZGSK**
> >
> > Dear Reviewer,
> >
> > Thanks again for your valuable comments! As the discussion period will end soon, could you please kindly check our responses and revisions? We believe that our responses and revisions have addressed some of your concerns.

---

> > > ### Comment · Reviewer_ZGSK · 2021-11-29
> > > **Thank the authors for the detailed reply**
> > >
> > > I would like to thank the authors for the detail reply to my comments. I understand now the separation the mixup process and the privacy protection. The additional experiments are valuable for showing the advantage of method.
> > > I have no further questions at this time point.

---

> > > > ### Author Response · Authors · 2021-12-01
> > > > **Response to Reviewer ZGSK**
> > > >
> > > > Dear Reviewer,
> > > >
> > > > Thank you for increasing your rating and for your response.
> > > >
> > > > Best regards,
> > > >
> > > > Authors

---

### Decision · Program_Chairs · 2022-01-20

**Decision:**

Reject

**Comment:**

A heterogeneous federated learning framework is proposed which does
not require auiliary public data sets, and does not reveal the private
data to the server or answering parties if they operate as
honest-but-curious entities. It builds a new protocol for private
inference, which can run on GPUs, and proposes a dataset expansion
method to not need an auxiliary data set. The paper presents extensive
empirical experiments on the method.

The paper was extensively discussed with the authors. The concerns
included both technical issues and more general issues on missing DP
guarantees and realisticness of the threat model. Many of the issues
were resolved by the clarifications provided by the authors, and as a
result two reviewers increased their scores. However, all reviewers still
place the paper to the borderline.

While the paper contains solid work, and improves efficiency compared to
previous models, this is a borderline paper where the final judgement
needs to be based on importance of the presented new contributions in
advancing the field. The paper may not yet quite reach the bar, but
I believe the reviewer comments have enabled the authors to improve the
paper for further work.